# Static and dynamic coding in distinct cell types during associative learning in the prefrontal cortex

Francesco Ceccarelli [1], Lorenzo Ferrucci [1], Fabrizio Londei [1,2], Surabhi Ramawat [1], Emiliano Brunamonti[1] & Aldo Genovesio [1] ✉

The prefrontal cortex maintains information in memory through static or dynamic population codes depending on task demands, but whether the population coding schemes used are learning-dependent and differ between cell types is currently unknown. We investigate the population coding properties and temporal stability of neurons recorded from male macaques in two mapping tasks during and after stimulus-response associative learning, and then we use a Strategy task with the same stimuli and responses as control. We identify a heterogeneous population coding for stimuli, responses, and novel associations: static for putative pyramidal cells and dynamic for putative interneurons that show the strongest selectivity for all the variables. The population coding of learned associations shows overall the highest stability driven by cell types, with interneurons changing from dynamic to static coding after successful learning. The results support that prefrontal microcircuitry expresses mixed population coding governed by cell types and changes its stability during associative learning.

Adaptive behavior requires the ability to generate an associative mapping between sensory information and the correct behavioral responses to achieve a goal. Depending on how familiar the context is, such a process may require either learning new associations or recalling those already established. The prefrontal cortex (PFC) is thought to be essential in the generation of associative mechanisms in goal-directed behavior[1]. Several neurophysiological studies have shown that cells in the PFC encode sensory cues[2,3], motor responses[4–6], abstract rules[7,8], and stimulus-response (S-R) associations[9,10].

Considerable effort has been made to investigate how the PFC maintains the representation of such task-related information over time[11–16] through a persistent[2,11,17–21] or a transient and flexible selectivity[22–25] resulting in either a static[15,26,27] or dynamic[28,29] coding scheme at the population level. Another line of research has focused on investigating the role of putative pyramidal cells and interneurons in task-related information processing in the PFC[3,30–38]. Pyramidal glutamatergic cells and GABAergic interneurons constitute the building blocks of cortical microcircuitry[39,40], providing local mechanisms for processing information in the PFC and for projecting such information to external areas in the case of pyramidal neurons[32]. Converging evidence from intracellular and extracellular recordings, coupled with the morphological characterization of the cells, indicates a consistent dissociation in firing patterns and action potential duration between cell types in different animal models[41–46], with pyramidal cells characterized by low firing rates and broad action potentials, and interneurons by narrow action potentials and high firing rates.

Despite the clear role of PFC in associative learning, the contribution of distinct cell types to associative learning and how the plastic adaptation induced by extended learning affects the microcircuitry are still an open question. Specifically, how S-R associations or mappings are established and maintained at the local microcircuitry level and the temporal dynamics of their coding during learning and after such associations are consolidated by familiarity have not yet

[1]Department of Physiology and Pharmacology, Sapienza University, 00185 Rome, Italy. [2]PhD program in Behavioral Neuroscience, Sapienza University, Rome, Italy. ✉e-mail: aldo.genovesio@uniroma1.it

been investigated. Similarly, the contribution of distinct cell types in driving the magnitude of associative coding in PFC microcircuitry remains uncertain, since conflicting evidence has been presented while studying different task-related information in PFC and other cortical areas[31–34,36,47].

Our previously acquired datasets in the PFC with a novel (Novel-Map) mapping task, a familiar (FamMap) mapping task, and a Strategy task[7] (Fig. 1a, b) are ideal for addressing the study of such coding properties of the local microcircuitry. In the NovelMap and FamMap tasks, monkeys were required to map associations between three

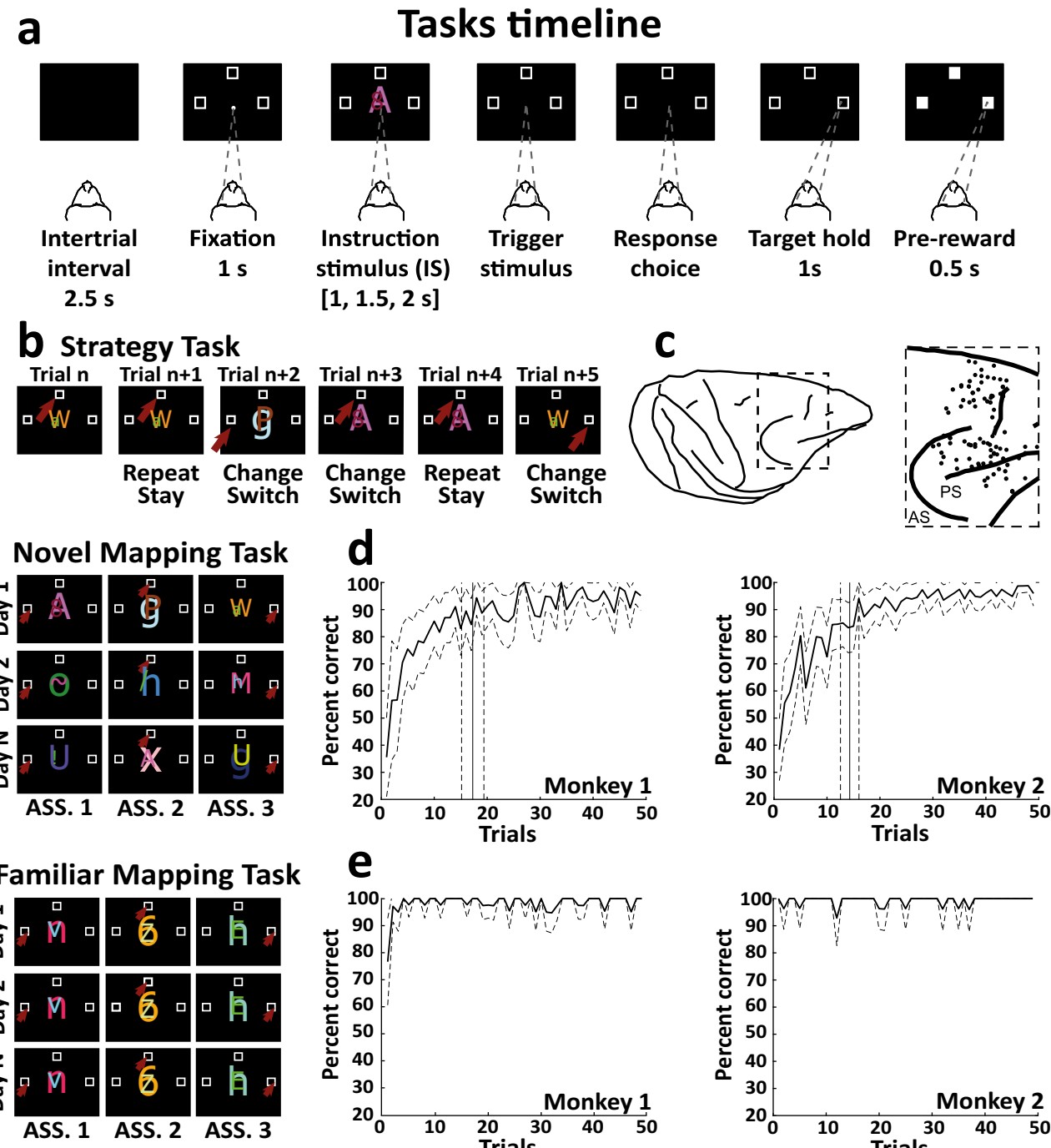

**Fig. 1 | Experimental tasks, recording locations and learning curves. a** The sequence of events in common between the three experimental tasks. Black squares represent the video screen, the white dot indicates the fixation spot, the empty white squares show the response targets, and the dotted lines represent the gaze of the monkey on the chosen target. **b** Examples of instructor stimulus sets used in the recording sessions in FamMap unchanged throughout the recording days, NovelMap, and Strategy task with the stimulus sets changed in each recording session and shared between the two tasks. A sequence of trials is shown in the Strategy task, and the two classes of trials and strategies (repeat and change trial, stay and switch strategy) of the task. Red arrows indicate the correct response for that trial. **c** Penetration sites of the recordings in the two monkeys. Percentage of correct choices in the initial 50 trials (second to 50th trial) of the NovelMap (**d**) and FamMap (**e**) task sessions divided by each male monkey. Dashed lines enclosing the curves indicate the 95% confidence interval (CI) limits. The vertical solid and dashed lines (**d**) correspond to the mean trials of learning completion (see Methods) and ±1 standard error of the mean (SEM), respectively. AS Arcuate sulcus, PS Principal sulcus, ASS. Stimulus-response association. Source data are provided as a Source Data file.

instruction stimuli (IS) and three response targets. In the NovelMap sessions, they were required to learn the associations for a new set of stimuli each time, whereas in the FamMap sessions, highly familiar ISs were presented. In the Strategy task, monkeys were trained to use two abstract response strategies: a repeat-stay strategy that required staying with the previous response when the same IS was repeated, and a change-shift strategy requiring a response shift when the IS changed from the previous trial. We used the Strategy task to study an additional task context not involving learning, and to investigate the coding properties of responses and stimuli separately because no association was required for the task.

Recent studies have demonstrated the copresence of both static and dynamic population coding[48,49] associated with distinct neuronal ensembles[50] and differentiated by functional discharge properties (i.e., timescales)[51,52] at the population level in the PFC. Such studies have helped to move away from a dichotomous view of populations characterized by only fixed static or dynamic coding mechanisms by suggesting their mixed contribution to the local microcircuitry of the PFC.

What has not yet been determined is whether such coding schemes are associated with the activity of specific cell types. Here, we find that both novel S-R associations in the learning context of the NovelMap task, and responses and stimuli in the Strategy task are encoded with a mixed coding scheme in the PFC: strongly static by putative pyramidal cells and dynamic by interneurons. Finally, we observe a general increase in stability in the presence of well-established associations in the FamMap and after the learning phase of the NovelMap task driven primarily by a flexible change in the coding scheme of interneurons from dynamic to static.

## Results

### Behavioral results
In the Strategy task, both monkeys performed accurately overall in repeat and change trials, with an average performance of 98.7% percent and 97.8% corrected trials, respectively. We applied a moving average method[53] (see Methods) to identify the trial when learning was completed in the NovelMap task. We found that both monkeys quickly learned the three S-R associations (Fig. 1d), on average within $16 \pm 0.7$ trials (monkey 1: $17.3 \pm 1.1$ trials, monkey 2: $14.3 \pm 0.9$ trials, two-sided Wilcoxon rank-sum test between monkeys $p = 0.23$), with an average overall performance of 93.0% correct responses over all trials (monkey 1: 93.6%, monkey 2: 92.4%). In the FamMap task, the monkeys with familiar associations performed almost perfectly (Fig. 1e), with an average performance of 98.6% (monkey 1: 98.0%, monkey 2: 99.1%).

### Waveform classification results
Our database included 1457 single units (monkey 1: 816; monkey 2: 641) after an initial preselection (see Methods) that were recorded in at least one (cells recorded in one task: 575, two tasks: 482, three tasks: 400) of the three tasks (Strategy task: 1306, 725 from monkey 1 and 581 from monkey 2; NovelMap task: 881, 514 from monkey 1 and 367 from monkey 2; FamMap task: 552, 275 from monkey 1 and 277 from monkey 2). The recordings were made in the dorsolateral and dorsomedial prefrontal cortex (Fig. 1c), mainly extending to area 46 and dorsal area 9[7].

To classify the cells in our database as narrow waveforms spiking (NW) putative interneurons and broad waveforms spiking (BW) putative pyramidal cells, we first calculated the trough-to-peak duration from the interpolated mean waveforms, and a two Gaussian mixed model was fitted on the obtained trough-to-peak distribution (see Methods). To increase the classification's statistical power, we pooled together data from all the three tasks[47]. The Hartigan dip test confirmed the bimodality of the trough-to-peak distribution (original Hartigan dip test $p = 0.0021$; calibrated Hartigan dip test $p < 0.001$), and both Akaike's and Bayesian information criteria indices were reduced (from $-2357$ to $-3043$ and from $-2346$ to $-3016$, respectively)

by performing a fit with the two-Gaussian model with respect to a one-Gaussian model. Such a best fit justifies the use of a two-Gaussian model[38,47,54]. Figure 2a shows the trough-to-peak distribution and mean waveforms of populations classified as BW and NW in the overall database. Our classification approach allowed us to classify 987 (67.7%) cells as BW and 332 (22.8%) cells as NW, while 138 (9.5%) cells were left unclassified (NC) due to their position within the distribution affecting the area between the two cutoffs of the Gaussian fits (see Methods)[38]. The classification of cell types was robust in all tasks (Table 1), and the proportions did not differ significantly between the classifications performed with the entire database (Chi-square test: BW pooled tasks vs BW Strategy task, $p = 0.99$; BW pooled tasks vs BW NovelMap task, $p = 0.34$; BW pooled tasks vs BW FamMap task, $p = 0.87$; NW pooled tasks vs NW Strategy task, $p = 0.82$; NW pooled tasks vs NW NovelMap task, $p = 0.55$; NW pooled tasks vs NW FamMap task, $p = 0.84$; NC pooled tasks vs NC Strategy task, $p = 0.77$; NC pooled tasks vs NC NovelMap task, $p = 0.50$; NC pooled tasks vs NC FamMap task, $p = 0.59$).

Several previous studies have categorized BW and NW cells in relation to their different intrinsic firing properties as regular and fast-spiking neurons[35,36,41,43,47]. Taking these previous studies into consideration in order to further validate our classification, we analyzed three firing metrics during the fixation period (0.0–1.0 s; see Fig. 1a for the task timeline), which we considered as our baseline period, in order to investigate the intrinsic firing properties of cell types. We calculated the Fano factor, mean firing rate, and the coefficient of variation of the interspike interval distribution (CV) for each neuron during this period (see Methods). Figure 2b–d shows the three calculated metrics. Consistent with previous studies, NW cells showed a higher average firing rate than BW cells ($p = 6.7 \times 10^{-15}$, two-sided Wilcoxon rank-sum test). BW cells also exhibited lower spiking variability, suggesting more regular firing activity over time than NW cells, as shown by the Fano factor ($p = 5.2 \times 10^{-19}$, two-sided Wilcoxon rank-sum test) and CV ($p = 1.4 \times 10^{-12}$, two-sided Wilcoxon rank-sum test).

### Population results
We investigated the contribution of cell types and the magnitude of their selectivity, measured as the percentage of explained variance ($\omega^2$), for novel and familiar S-R associations in the NovelMap and FamMap tasks, and for both response and stimulus in the Strategy task. For novel and familiar S-R associations in the NovelMap and FamMap tasks, we analyzed only correct trials where each of the three stimuli was associated with only one response following the S-R mappings defined by the specific task rule. Associations between stimuli and responses were not changed during the mapping sessions. Consequently, we did not distinguish the contribution of response and stimulus cell selectivity to the coding of the S-R associations. However, as shown later, we used the Strategy task to study the coding of different cell types for stimulus and response independently. For the response and the stimulus in the Strategy task, we selected the correct trials in which the response corresponded to one of the three possible correct target positions (top, right, left) and the three stimuli displayed, respectively. We compared the average $\omega^2$ across the BW and NW neurons to evaluate the amount of variability in neuronal activity explained by the two types of cells in encoding the task variables during the time course of the trial (see Methods). We considered the IS period as the epoch of interest for all subsequent analyses, which includes the period from the IS onset to 1 second after it (see Fig. 1a for the task timeline). Cells recorded for less than 10 correct trials in each task variable condition (each variable of interest had 3 conditions) were excluded from this analysis. Our final database of cells in the associative tasks that we used for the analysis included 487/580 BW and 190/210 NW cells recorded in the NovelMap task and 275/376 BW and 98/128 NW cells in the FamMap task. Our final database of cells in the Strategy task included 674/885

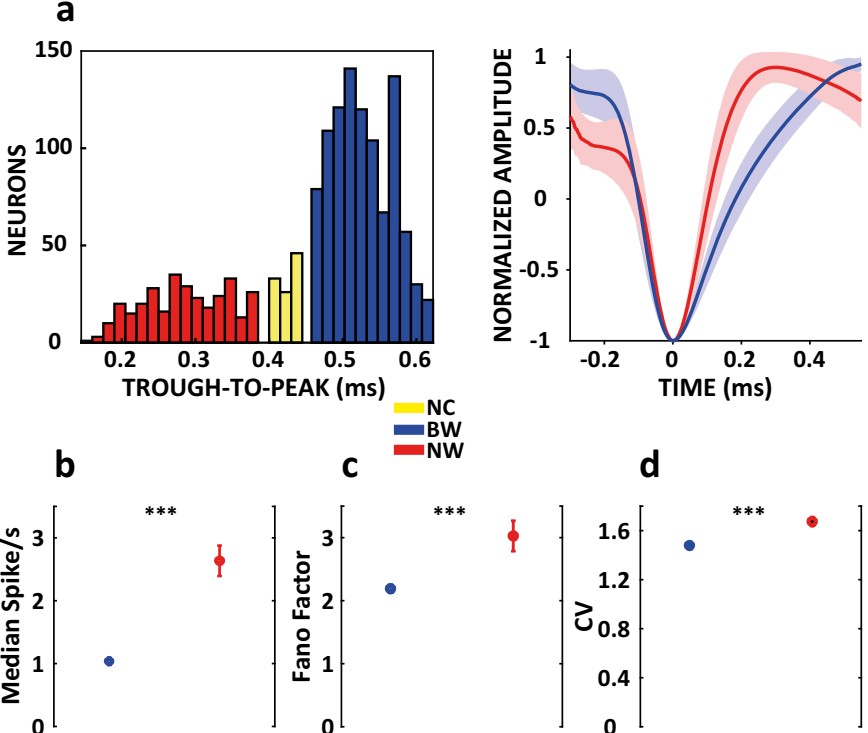

**Fig. 2 | Cell types classification. a** Complete database of the waveforms trough-to-peak duration distribution in milliseconds (left). Colors indicate cells classified as BW (blue), NW (red), and NC (yellow); normalized mean interpolated waveforms of BW and NW populations (right), shaded areas represent standard deviation (SD). **b**–**d** Firing metrics calculated separately for BW and NW populations during the baseline period of fixation (0-1 s): firing rate (**b**), Fano factor (**c**) and coefficient of variation (CV) (**d**). Colored circles indicate the median values and error bars show the standard error of mean (SEM) (BW: 778 cells; NW: 274 cells, with at least 30 completed trials). Two-sided Wilcoxon rank-sum test: ***$p < 0.001$ (**b**), $p = 6.7 \times 10^{-15}$; (**c**), $p = 5.2 \times 10^{-19}$; (**d**), $p = 1.4 \times 10^{-12}$). Source data are provided as a Source Data file.

BW and 232/293 NW cells for the stimulus and 664/885 BW and 226/293 NW cells for the response.

The analysis of the amount of variability explained by both the population of BW and NW cells revealed that both groups of neurons significantly encoded the S-R associations in both the NovelMap (cluster-based permutation test, $p < 0.001$) and FamMap (cluster-based permutation test, $p < 0.001$) tasks throughout the duration of the IS period (Fig. 3a, b, blue and red bars, for BW and NW populations, respectively). Interestingly, the coding in the two tasks differed for latency and peak of activity between the two populations. In the NW population, the associations were coded with shorter population latency and a clear peak in the early phase of the IS period than in the BW population. To test for a difference in the selectivity latencies, we identified the first significant time bin within the IS period in the two cell type populations. We found that NW population coded significantly earlier, both novel (NovelMap, mean onset time, from IS onset: NW 226.9 ms; BW 289.7 ms, Kruskal-Wallis between time onset distributions, $p = 0.024$) and familiar associations (FamMap, mean onset time, from IS onset: NW 179.1 ms; BW 265.4 ms, Kruskal-Wallis between time onset distributions, $p = 0.016$). Moreover, we found a significant decrease in latencies of familiar versus novel associations

coding, in the NW population (Kruskal-Wallis between NW time onset tasks distributions, from IS onset, $p = 0.034$), but not in the BW population (Kruskal-Wallis between BW time onset tasks distributions, from IS onset, $p = 0.40$). In addition, the coding of the associations decreased in the late phase of the IS period in the NW but not in the BW population, which was instead characterized by a more constant coding throughout the IS period. In the NovelMap task, the NW population selectivity was higher than the BW population throughout the early IS and part of the late IS period (cluster-based permutation test, $p < 0.001$; first bin time: 160 ms, last bin time: 745 ms, from IS onset). In contrast, in the FamMap task, such higher selectivity for the NW population was limited to the early phase of the IS period (cluster-based permutation test, $p < 0.001$; first bin time: 130 ms, last bin time: 415 ms, from IS onset) (Fig. 3a, b, black bars). To further investigate the engagement of BW and NW populations in associative coding, we recomputed $\omega^2$ in two time bins reflecting the two phases of the IS period and identified the cells modulated significantly (early: 50–450 ms and late: 500–900 ms IS period, cluster-based permutation test, $p < 0.05$). As shown by the histograms in Fig. 3a, b, we found an overall larger engagement of the NW population than the BW population, measured in terms of percentages of cells modulated by the S-R associations (NovelMap early IS period, NW: 32%; BW: 16%; NovelMap late IS period, NW: 31%; BW: 17%; FamMap early IS period, NW: 32%; BW: 14%; FamMap late IS period, NW: 21%; BW: 19%). Explained variance cells' distributions in the investigated sample for novel and familiar associations confirmed an important involvement of both cell type populations in the IS early period and the highest population coding of the NW population (Supplementary Fig. 1a, b).

To further study the distinct cell types' contribution in the associative coding during the learning and the post-learning periods, where

**Table 1 | Number and percentage of cells classified for each experimental task**

| Task | Broad cells (%) | Narrow cells (%) | Unclassified (%) |
|---|---|---|---|
| Strategy | 885 (67.8%) | 293 (22.4%) | 128 (9.8%) |
| NovelMap | 580 (65.8%) | 210 (23.8%) | 91 (10.3%) |
| FamMap | 376 (68.1%) | 128 (23.2%) | 48 (8.7%) |

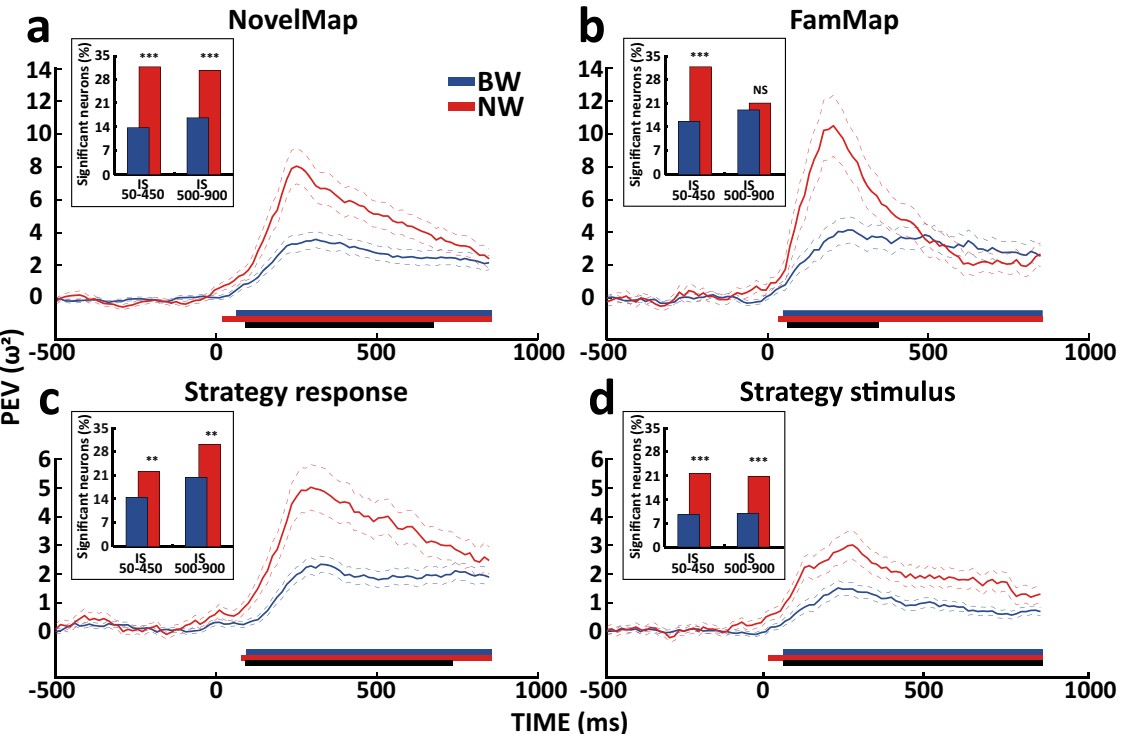

**Fig. 3 | Cell types explained variance and percentages of selective cells for novel and familiar associations, stimulus and response.** Percentage of explained variance ($\omega^2$) averaged on the BW (blue) and NW (red) populations for the associations coding in the NovelMap (**a**), FamMap (**b**), responses (**c**) and stimuli (**d**) in the Strategy task. Data aligned to the IS onset, covering 1 s of the IS period and the last 500 ms of the fixation period. Colored dashed lines show ±1 SEM of $\omega^2$. Colored bars below the figures mark $\omega^2$ values significantly higher than the null distribution (cluster-based permutation test $p < 0.001$, see Methods); black bars highlight a significant difference between the two populations (cluster-based permutation test $p < 0.001$, see Methods). Histograms within each figure display the percentages of significant cells for BW and NW populations in the early (50–450 ms) and late (500–900 ms) phases of the IS period. Chi-square test: **$p < 0.01$; ***$p < 0.001$; NS: not significant (**a**) 50–450, $p = 9.4 \times 10^{-8}$, 500–900, $p = 5.7 \times 10^{-5}$; (**b**) 50–450, $p = 6.5 \times 10^{-4}$, 500–900, $p = 0.63$; (**c**) 50–450, $p = 0.007$, 500–900, $p = 0.002$; (**d**) 50–450, $p = 1.7 \times 10^{-6}$, 500–900, $p = 1.5 \times 10^{-5}$). Source data are provided as a Source Data file.

associations are consolidated, we identified in each NovelMap session the trial when learning was completed and accordingly split the recorded sessions into two blocks of trials: the learning (from the first trial to the learning completion trial) and the post-learning block (from the trial after the learning completion trial to the last one recorded in the session). In this analysis, we selected the cells from the previously analyzed sample with at least 6 trials for each association in the learning (NovelMap learning block, NW: 42/210; BW: 87/580) and post-learning blocks (NovelMap post-learning block, NW: 152/210; BW: 376/580). The results observed mirrored the selectivity profiles found between NovelMap and FamMap using the whole session (Supplementary Fig. 2a–d). Populations of both the cell types encoded the associations over the entire IS period in both the learning and post-learning blocks (cluster-based permutation test, $p < 0.001$). In the learning block, the NW population encoded the associations stronger than the BW population for most of the IS period (cluster-based permutation test, $p < 0.001$; first bin time: 220 ms, last bin time: 775 ms, from IS onset). However, in the post-learning block, the stronger selectivity, was restricted mostly to the early IS period (cluster-based permutation test, $p < 0.001$; first bin time: 70 ms, last bin time: 595 ms, from IS onset), to vanish later.

Next, we investigated whether the stronger coding of the task variables displayed by the NW population could be generalized to the non-associative context of the Strategy task. In this task, each stimulus is not associated with any fixed response, since the response depends on the strategy used (Fig. 1b, Strategy task trial sequence). Because of this, the Strategy task allows us to dissociate stimulus and response signals, which is not possible in the associative tasks. Similar to the previous results, even in the Strategy task, the NW population was

more selective than the BW population for coding both the response (Fig. 3c; cluster-based permutation test, $p < 0.001$; first bin time: 160 ms, last bin time: 805 ms, from IS onset) and the stimulus information (Fig. 3d; cluster-based permutation test, $p < 0.001$; first bin time: 115 ms, last bin time: 925 ms, from IS onset). We found, as for the associative coding, a significantly greater percentage of cells encoding response and stimulus (Fig. 3c and 3d, histograms) in the NW than in the BW population (Strategy response early IS period, NW: 22%; BW: 14%; Strategy response late IS period, NW: 30%; BW: 20%; Strategy stimulus early IS period, NW: 22%; BW: 9%; Strategy stimulus late IS period, NW: 21%; BW: 10%). The distribution of the explained variance confirms the involvement of both cell types along with a higher selectivity of the NW population (Supplementary Fig. 1c, d). The NW population encoding of the response peaked in the early phase of the IS period and progressively decreased in its late phase, in contrast to a more constant coding of the BW population. The NW population encoding of the stimulus reached the maximum coding capacity in the early IS period, decreasing thereafter with a similar temporal trend as in the BW population. Latency analyses revealed a tendency in the NW population to encode both the response (Strategy response, mean onset time, from IS onset: NW 289.3 ms; BW 327.1 ms, Kruskal-Wallis between time onset distributions, $p = 0.084$) and the stimulus (Strategy stimulus, mean onset time, from IS onset: NW 229.2 ms; BW 283.5 ms, Kruskal-Wallis between time onset distributions, $p = 0.21$) earlier in the Strategy task, when compared to the BW population, however without achieving statistical significance.

Finally, we assessed the impact of some additional factors that might affect the magnitude of coding. To test for a possible confound from the firing rate differences between BW and NW populations on

the observed differences in the coding magnitude, we recomputed the average firing rate during the 1-s fixation period for each task and variable analyzed. As expected, the results confirmed the inherent difference in firing activity between BW and NW populations for stimulus ($2.1 \pm 0.1$ sp/s, $5.1 \pm 0.4$ sp/s) and response ($2.2 \pm 0.1$ sp/s, $5.1 \pm 0.4$ sp/s) in the Strategy task and for novel ($2.1 \pm 0.1$ sp/s, $5.0 \pm 0.4$ sp/s) and familiar ($2.1 \pm 0.1$ sp/s, $4.4 \pm 0.5$ sp/s) associations in NovelMap and FamMap, respectively. We further matched the firing rates to rule out the possibility that neurons with high firing rates were responsible for the differences in coding strength. We sorted neurons in the NW population by firing rate, and then we removed cells from the population (44 for response and stimulus Strategy task, 39 novel associations NovelMap, 15 familiar associations FamMap) until the mean rate was lowered[5] to $2.5 \pm 0.2$ sp/s, making the firing rates comparable to those in the BW population. We then repeated the $\omega^2$ analysis, confirming the results shown in Fig. 3. For each variable, the coding was significant in the whole IS period, as it was the greater coding of the NW population (Strategy response, cluster-based permutation test, $p < 0.001$; first bin time: 145 ms, last bin time: 715 ms; Strategy stimulus, $p < 0.001$, 115 ms, 880 ms; NovelMap, $p < 0.001$, 160 ms, 580 ms; FamMap, $p < 0.001$, 130 ms, 385 ms, from IS onset) (Supplementary Fig. 3a, d, g, j, black bars). We next assessed whether the presence of cells not modulated by the tasks reduced the observed $\omega^2$ values. We defined neurons as task-related if their firing rate in the baseline period (fixation period: 0.0-1.0 s) differed significantly (Kruskal-Wallis, $p < 0.05$) from any of the two phases (early IS period: 50–400 ms; late IS period: 400–900 ms) of the IS period (Strategy response, BW 567/664 and NW 200/226; Strategy stimulus, BW 574/674 and NW 203/232; NovelMap, BW 377/487 and NW 151/190; FamMap, BW 201/275 and NW 84/98). We found that overall, the BW and NW task-related populations had higher $\omega^2$ values and a higher coding magnitude in the NW population, comparable to the results in Fig. 3 (Supplementary Fig. 4a, d, g, j). Finally, we tested whether the results were affected by cells with low firing rate. For this analysis, we eliminated the cells with a firing rate less than or equal to 0.5 Hz throughout the recording session (Strategy response, BW 475/664 and NW 192/226; Strategy stimulus, BW 481/674 and NW 195/232; NovelMap, BW 350/487 and NW 160/190; FamMap, BW 203/275 and NW 81/98). Such selection criterion also did not result in any change in the previously observed effects (Supplementary Fig. 5a, d, g, j).

## Cross-temporal decoding results

After evaluating how both cell type populations encoded the task variables in terms of strength and latency, we examined how the representation of these variables evolved over time during the IS period. For this purpose, we performed a cross-temporal decoding analysis in which we trained and tested a linear classifier using all the possible pairs of time bins[28,29], limiting the analysis to the IS period. This approach produces a classification accuracy matrix (i.e., the ability of the classifier to discriminate the conditions of the variable under consideration), in which the off-diagonal values allow us to evaluate the similarity of the representation across the time bins when directly compared to the corresponding on-diagonal values (i.e., the time bins used to train and test the classifier in the specific off-diagonal data point)[26,55,56], according to a dynamic or static scheme[57]. In this context, if the classifier's performance would not be affected using different time bins, the coding scheme can be defined as static over time. Otherwise, if the classifier's performance is reduced, remaining constant only for time bins close in time, the coding scheme can be defined as dynamic.

We quantified and statistically tested the difference between the on- and off-diagonal prediction accuracies of the classifier by comparing the off-diagonal prediction accuracies with the predictions obtained when the training and test bins were identical, that are the on-diagonal values (see Methods). Figures 4 and 5 show the cross-

temporal decoding normalized matrices and the associated classification matrices of the static off-diagonal data points for the associations in the NovelMap (Fig. 4a, b) and FamMap (Fig. 4c, d) tasks and for the response (Fig. 5a, b) and stimulus (Fig. 5c, d) in the Strategy task. In addition, Supplementary Fig. 6 shows non-normalized cross-temporal decoding matrices. We observed a general tendency of the BW population to express a higher percentage of off-diagonal data points classified as static than the NW population in all the tasks and variables analyzed (Strategy response, BW: 34.7%, NW: 12.5%; Strategy stimulus, BW: 35.2%, NW: 11.7%; NovelMap associations, BW: 27.2%, NW: 13.7%; FamMap, BW: 41.2%, NW: 24.2%; Chi-square test all comparison, $p < 0.001$).

To investigate the temporal evolution of neural coding stability more quantitatively, we compared the BW and NW populations by implementing a stability index that estimates the proportion of off-diagonal data points classified as static[26,55,56] (see Methods). To ensure a fair comparison, we applied a number-matching procedure to rule out that the numerosity of the sample could affect the results, selecting an equal number of cells for BW and NW populations in each task (Strategy response: 200; Strategy stimulus: 200; novel association NovelMap: 165; familiar associations FamMap: 80). We observed a significant difference in the population coding scheme as reflected by the coding stability in the IS period between the NW and BW populations for the novel associations in the NovelMap task (cluster-based permutation test, $p < 0.001$). The NW population showed significantly lower stability than the BW population (Fig. 4e), suggesting that a moderately dynamic coding scheme was maintained for the whole IS period. On the other hand, the BW population was characterized by a progressive increase in stability in the early phase of the IS period, reaching the peak at ~625 ms after the presentation of the IS but still maintaining relatively high stability in the later IS period. Familiar associations in the FamMap task (Fig. 4f) were represented through a static scheme in the BW population with a moderate increase in magnitude compared with the novel associations in the NovelMap task in the late IS period, although this increase was not statistically significant (Fig. 6a). However, we observed an earlier increase in stability starting from 150 ms up to the end of the early phase of the IS period that reached its peak at approximately 475 ms (Fig. 6a) in the FamMap compared to the NovelMap task (cluster-based permutation test, $p < 0.001$). Surprisingly, although we still found the NW population to be less static than the BW population (Fig. 4f) (cluster-based permutation test, $p < 0.001$), when comparing the two tasks, we observed a switch to a more static coding scheme in the late IS period but from the NovelMap to the FamMap task (Fig. 6b) with a significant increase in stability between the two tasks (cluster-based permutation test, $p < 0.001$). Static and dynamic coding schemes are thought to be the product of an interplay between neurons of the decoded population[50]. Single unit selectivity properties such as duration of selectivity[28,50,55] and preference switching[55] contribute to such coding schemes. Supplementary Fig. 7 shows examples of single units classified as BW in NovelMap (Supplementary Fig. 7a) and FamMap (Supplementary Fig. 7b). Each of these cells and the NW cell in FamMap (Supplementary Fig. 7b) exhibited sustained selectivity throughout the IS period associated with a stable coding scheme. Supplementary Fig. 7a shows a NW cell recorded in NovelMap, characterized by a transient selectivity that support a dynamic coding scheme.

We then repeated the analysis using the learning and post-learning blocks of trials within the NovelMap sessions. Interestingly, in the learning block (Supplementary Fig. 2b, c) the NW population showed a significant reduction of the levels of stability (cluster-based permutation test, $p < 0.001$), suggesting the use of a dynamic coding scheme during the IS period, compared to the high stability levels achieved in the late IS period by the BW population, indicating a purely static scheme. However, in the post-learning block (Supplementary Fig. 2e, f), we found an overall increase in stability comparable to that

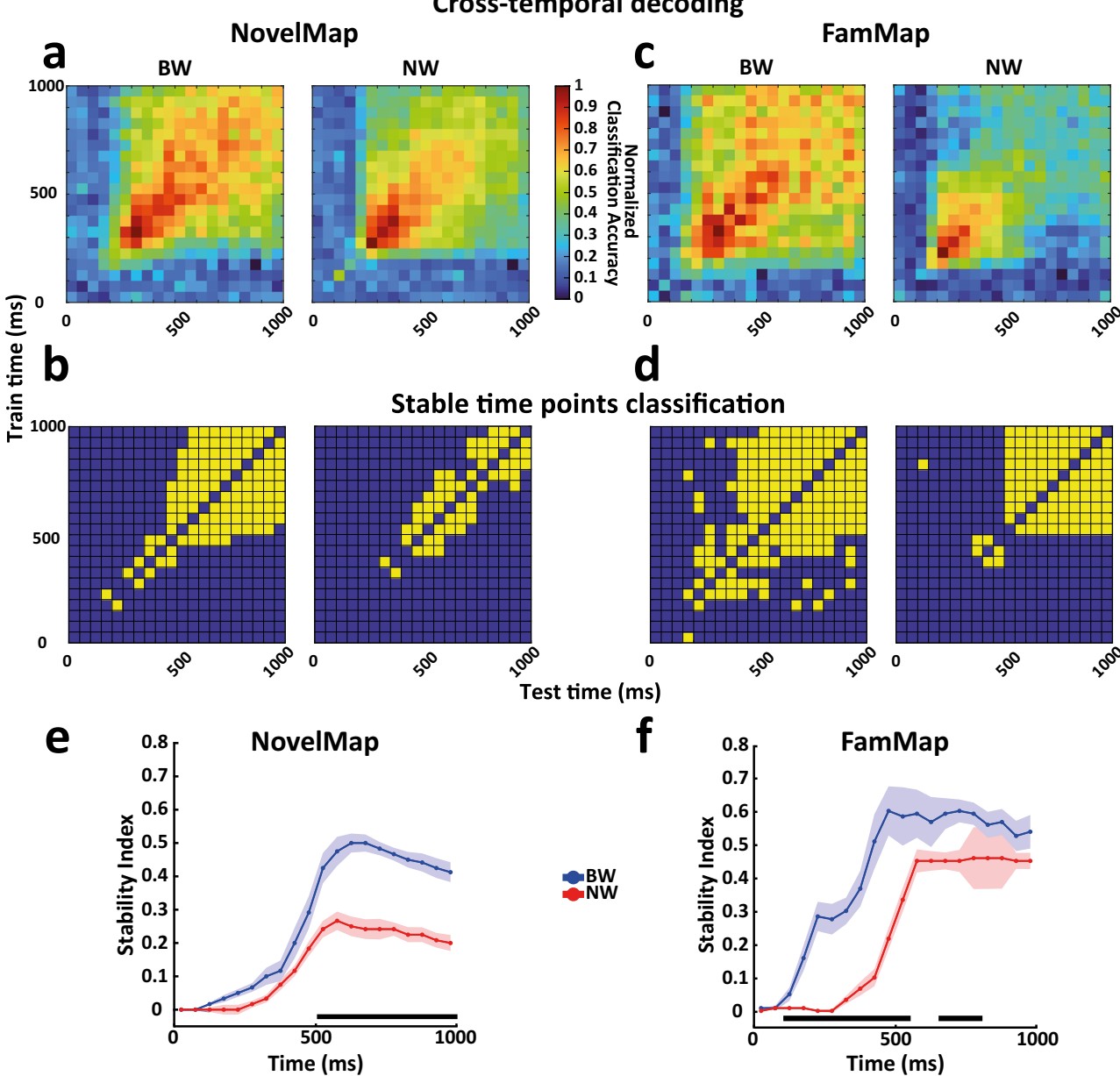

**Fig. 4 | Cross-temporal decoding and stability of cell types for novel and familiar associations.** Cross-temporal decoding, stable data point classification, and stability index for novel associations in NovelMap (**a**, **b**, **e**) and familiar associations in FamMap (**c**, **d**, **f**) for the BW and NW populations. The alignment refers to the presentation of the instructor stimulus. **a**, **c** Cross-temporal decoding, where the y-axis shows the time bins used for training and the x-axis shows the time bins used for testing the classifier. Normalized classification accuracy values are color-coded. **b**, **d** The maps show the classification of each data point (except for the on-diagonal data points, not considered in this analysis and left in blue by convention)

of the cross-temporal decoding as static (yellow) or not (blue) (see Methods). **e**, **f** Stability indices quantify the stability of the coding over time for each cell type (see Methods). High index values indicate strong stability. Shadow areas represent ±1 SD of indices computed independently by repeating decoding by resampling neurons with replacement for each cell type population (see Methods). Horizontal black lines show the time points with a significant difference in stability between the two populations (cluster-based permutation test $p < 0.001$). Source data are provided as a Source Data file.

reported in FamMap, where the BW population showed a significant rise in stability in the early phase of the IS period compared to the learning block (Supplementary Fig. 8a). The NW population in the post-learning block, although with lower levels of stability than the BW population (cluster-based permutation test, $p < 0.001$) (Supplementary Fig. 2e, f) exhibited a strong increment in stability compared to the previous learning block (Supplementary Fig. 8b). Such changes in cell types coding schemes are consistent with those found between the entire NovelMap and FamMap sessions, again suggesting a shift from a dynamic to a static coding scheme in the NW population even before

the associations become familiar. These results taken together suggest that such a coding scheme shift may occur fast after learning for novel mappings to be maintained thereafter.

Considering the Strategy task, both response (Fig. 5e) and stimulus (Fig. 5f) were represented similarly in BW and NW populations to what was observed for novel associations in the NovelMap task. In both cases, the NW populations were characterized by significantly lower stability (cluster-based permutation test, $p < 0.001$) during the whole IS period, suggesting a moderate dynamic scheme as in the NovelMap task. The BW populations increased the static levels in time,

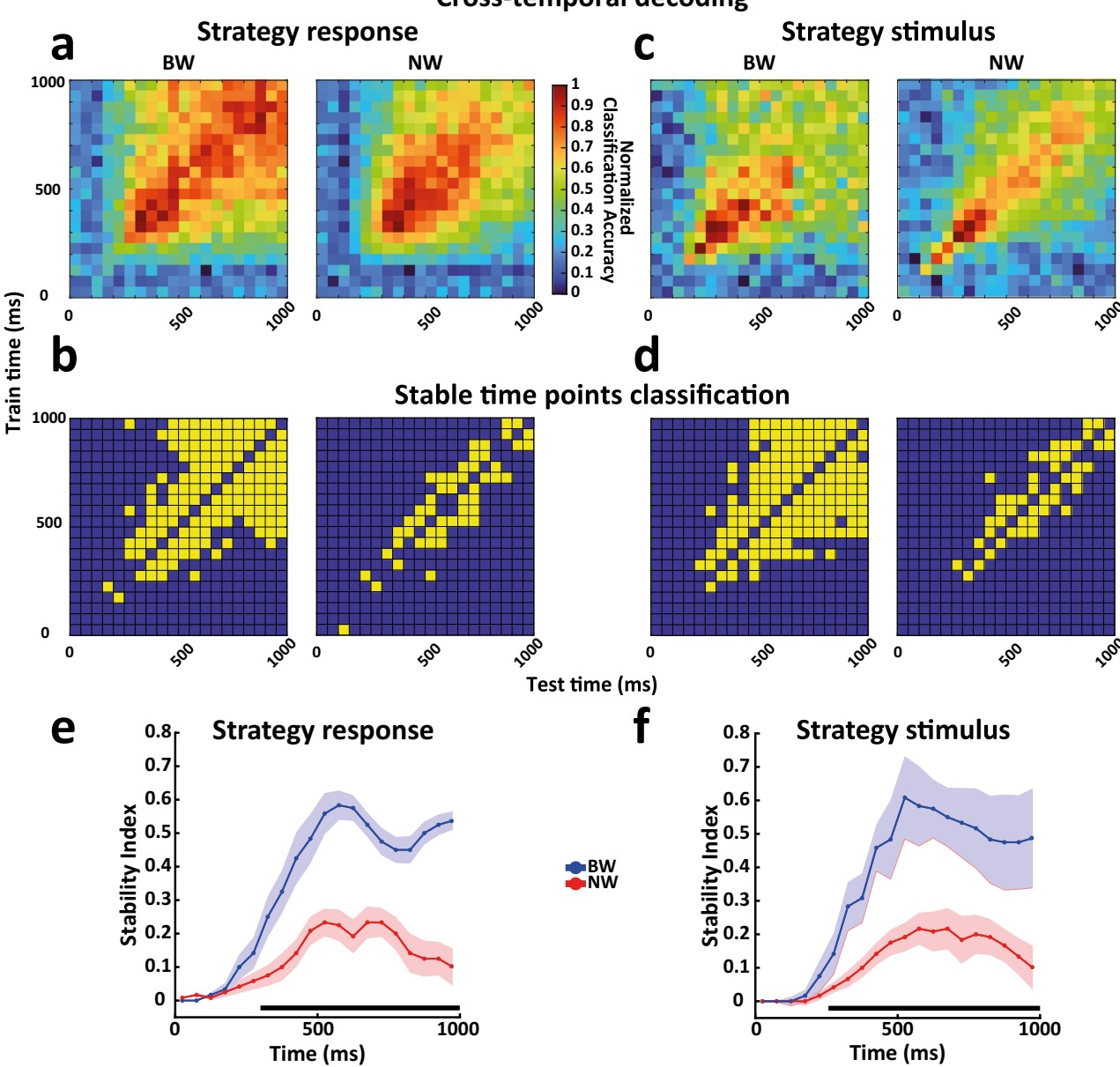

**Fig. 5 | Cross-temporal decoding and stability of cell types for stimulus and response.** Cross-temporal decoding, stable data point classification, and stability index for the response (**a**, **b**, **e**) and stimulus (**c**, **d**, **f**) in the Strategy task for the BW and NW populations. Shadow areas represent ±1 SD of indices computed independently by repeating decoding by resampling neurons with replacement for each cell type population (see Methods). The organization of the figure is the same as that in Fig. 4. Horizontal black lines show the time points with a significant difference in stability between the two populations (cluster-based permutation test $p < 0.001$). Source data are provided as a Source Data file.

albeit faster than in the NovelMap task, reaching their peaks at ~500 ms from the IS onset to stabilize only in the late IS period. Supplementary Fig. 7 shows two BW example cells, selective for response (Supplementary Fig. 7c) and stimulus (Supplementary Fig. 7d) in the Strategy task, whose persistent selectivity supports a static coding scheme at population level. Finally, the two NW example cells, with the first exhibiting a transient response selectivity (Supplementary Fig. 7c) and the second a stimulus preference switching (Supplementary Fig. 7d), indicate both a dynamic coding scheme.

We applied the control analyses used previously for the magnitude of selectivity to test the influence of factors that might impact the population coding schemes. To control that population coding schemes were maintained independently of differences in firing, we repeated the decoding analysis and stability index calculation using

the BW and NW populations with comparable firing rates, and we confirmed the results shown in Figs. 4, 5 and Fig. 6 (Supplementary Figs. 3 and 9a). Analogously, task-related BW and NW populations and populations with a firing rate greater than 0.5 Hz maintained a coding scheme consistent with those found in the main results (Supplementary Figs. 4, 5 and 9c, d, respectively). Finally, the main effects observed in NovMap and FamMap were also confirmed by matching the same sample of recorded cells using only the cells recorded in the same daily session in both tasks (Supplementary Figs. 10 and 9b).

## Discussion
In this study, we recorded single-unit activity in the macaque's dorsolateral and dorsomedial prefrontal cortex (PF) in two associative tasks, the NovelMap and the FamMap tasks, and in a Strategy

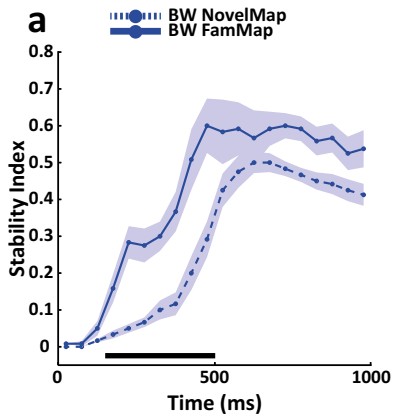
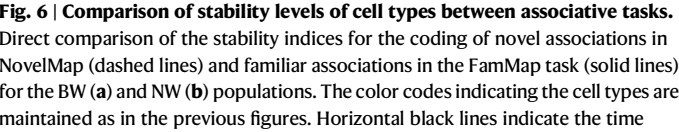
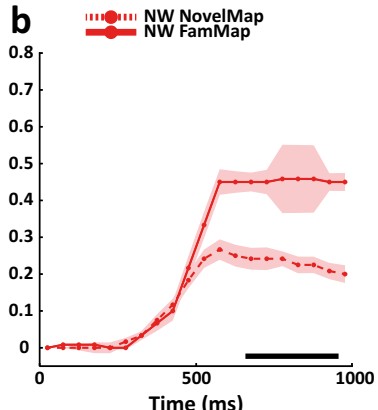

**Fig. 6 | Comparison of stability levels of cell types between associative tasks.** Direct comparison of the stability indices for the coding of novel associations in NovelMap (dashed lines) and familiar associations in the FamMap task (solid lines) for the BW (**a**) and NW (**b**) populations. The color codes indicating the cell types are maintained as in the previous figures. Horizontal black lines indicate the time

points with a significant difference in the stability levels between the two tasks (cluster-based permutation test $p < 0.001$). Shadow areas represent ±1 SD of indices computed independently by repeating decoding by resampling neurons with replacement for each cell type population (see Methods). Source data are provided as a Source Data file.

task. Previous studies using this and another simplified strategy-cued version of the task have shown that the PF is involved in multiple processes: the generation and maintenance of strategies[6,7,58], past and future response coding[4,6,59–61], stimulus-response associations[9,10], and choice and outcome monitoring[6], but the specific role of interneurons and pyramidal cells has not yet been investigated. In this study, we first focused on characterizing the neural coding of the S-R associations as static or dynamic in the populations of putative interneurons and pyramidal cells, both during learning and after learning. Second, we studied response and stimulus signals in the Strategy task to investigate how our findings generalized to a non-associative task.

To classify the cell types, we applied an unbiased clustering technique to the trough-to-peak duration of the cellular waveforms, dividing them into broad and narrow spiking cells. Previous studies have indicated this feature to be a good predictor for the identification of putative pyramidal cell and interneuron cell types[33,42,44,45,62], based at least partly on their differential expression of subtypes of sodium and potassium channels with different kinetic properties[63,64]. We found that the putative interneurons and pyramidal neurons represented 22.8% and 67.7% of the recorded cells, respectively, consistent with the proportions identified in previous PF studies in monkeys[3,31,33,35]. In addition, the two populations have been shown to have different firing metrics typical of fast and regular spiking cells, which are in turn associated with the intrinsic firing properties of interneurons and pyramidal cells[36,38,41,47,54,65].

In this study, we addressed the role of the different cell types. As expected from our previous study[7], we found that both cell types participated in the encoding of novel and familiar associations during the IS period. Comparing their neural selectivity and latency, we found that the putative interneurons were more selective in both tasks and at shorter latencies for the associations than the putative pyramidal neurons. Furthermore, by using the Strategy task, we were able to dissociate stimuli and responses, testing whether the results in the associative tasks could be generalized to both visual stimulus and response signals in a non-associative task. We found that, similarly to the associative tasks, both stimulus and response were coded more strongly and earlier by the putative interneurons. Overall, the percentage of variance explained by the task variables is consistent with that reported in previous studies in PFC[51,66,67]. Previous studies in PFC have found a similar stronger encoding of task-related information by interneurons than by pyramidal neurons in tasks involving numerical categorization[32] and reversal learning[36].

Our study extends to the learning context, demonstrating a key contribution of the putative prefrontal interneurons. However, this result cannot be taken as a general rule, and further studies with spatial oculomotor delayed-response[31] and pro/antisaccade tasks[33] did not show a difference in coding between cell types in the PF, while studies with direction and color discrimination[34,47] have shown increased encoding by pyramidal neurons. The presence of such discrepancy between studies indicates that the role of the interneurons can change based on the task requirement and context[3]. It has been hypothesized that the anticipation and, in some cases, stronger selectivity of putative interneurons may play a key role in refining and sharpening the representation of task-related information in cortical microcircuitry[39,40]. Putative interneurons and putative pyramidal neurons have shown divergence in their preferred activity[32], inverted tuning curves[32,68], and opposite synchronous firing patterns[30]. These studies suggest a role of putative interneurons in the selective inhibition of pyramidal neurons, sharpening their selectivity by suppressing their activity under conditions different from the preferred condition[68], leading to optimization of the circuitry in information processing.

A debate is still ongoing concerning the mechanisms needed to maintain task-related information in memory through delay periods. At first, the dominant view was that the maintenance of information was conveyed by a persistent sustained activity by the cells involved in information coding[17,18,26,27,69,70], establishing a static view of the representation conveyed at the population level as the hallmark of short-term memory. Such persistent activity has also been observed in relation to several other cognitive processes in addition to short-term memory[13]. However, further evidence has challenged this view by demonstrating a more dynamic coding than previously thought[23,24,28,29,71]. More recent studies have also shown that these coding schemes are not even mutually exclusive in a given brain area, but rather that they can coexist[48] and be associated with distinct neuronal populations[50–52], undermining the idea of rigid categorical separation and laying the foundations for conceiving mixed and heterogeneous representations in the prefrontal cortex. However, these recent studies lacked cell type classification, except in functional terms as linked to the intrinsic temporal properties of discharge in baseline periods[51,52], and did not investigate whether these differences depend on cell types.

Here, we asked whether these heterogeneous representations could be associated with different processing properties of the cell types. To address this question, we used a statistical evaluation

method of the off-diagonal reduction of accuracy to quantify coding schemes. The presence of an off-diagonal reduction indicates a certain degree of independence between activity patterns[16,57] and allows for the study of this coding property in different animal models[26,55,72] and humans[56,73]. Our method considers the coding schemes not categorically but as a continuum between stability and dynamism aiming to capture the high variability of the activity patterns described in the delay periods[74], particularly in PFC[22,25]. We tested whether the two cell classes differed in terms of stability of their neural coding by characterizing their temporal coding properties during the IS period. In this period, based on the strategy implemented, a response was planned and maintained in memory as in a typical delayed response task but without turning off the visual response targets. The three squares are not considered to be a visual cue because they do not provide information about the correct target being presented.

We observed a heterogeneous coding scheme for response and stimulus in the Strategy task and for the novel associations in Novel-Map strictly related to the identification of cell types. We found that in the late phase of IS period, the population activity was characterized by a considerably static coding scheme of putative pyramidal neurons as opposed to the continuous, moderately dynamic scheme of putative interneurons consistent in time throughout the IS period.

The heterogeneous coding schemes observed in our study present an opportunity to evaluate the short-term memory mechanisms proposed by recurrent network models. Our findings reveal that while the coding of putative pyramidal neuron populations for stimulus, response, and novel associations reached high levels of stability in the late IS phase, they also exhibited moderate dynamism in the early IS phase. Previous studies[51,52,55,75] that used cross-temporal decoding in PF found similar dynamics which are inconsistent with classical stable attractor models[76]. Unlike previous studies that did not differentiate between cell types, our findings report this effect in a specific population of putative pyramidal cells.

Previous computational and experimental studies identified two mechanisms of information maintenance in prefrontal cortex short-term memory: a static coding scheme and a dynamic coding scheme[11,12,74,77]. Static coding is believed to result from reverberating connections between pyramidal neurons in PF that are mediated by NMDA receptors[78,79], enabling a sustained representation during the delay before the action is performed. On the other hand, other studies have shown that information maintenance can occur without static coding, via transient strengthening of active synapses in the microcircuitry[80,81]. In this way, a temporary synaptic memory trace is formed through activity-dependent short-term synaptic plasticity, a process supported by dynamic coding[16,75]. The presence of heterogeneous coding schemes in the prefrontal cortex in our study supports the idea that recurrent network models, such as those proposed for reservoir computing[82], which can express both static and dynamic properties cohesively[83–85] may provide a useful model for the microcircuitry of the prefrontal cortex. Nevertheless, further studies will be needed to clarify the contribution of cell types within the framework of these models in the expression of these coding schemes.

Understanding how the microcircuitry is organized during extended learning remains a crucial issue in the field of associative learning[86,87]. A recent computational work by Barack et al.[88] challenged the view that during an early stage of task learning, a stable attractor model and its associated high stability could account for the dynamism observed in the prefrontal cortex. Instead, through neural network simulations, they suggest that the gradual strengthening of microcircuitry connections and structure over time may lead to increased population stability after extended learning[88]. Even though this model likely explains the dynamical states followed by the prefrontal cortex during different stages of learning, it needs to be backed by further experimental evidence[87]. By comparing novel and familiar

associations, we tested the hypothesis that stability increases during the representation of S-R associations. Our results indicate a learning-related effect on the coding stability of associations, with an overall more static representation in FamMap than in NovelMap, as proposed by Barak et al.[84], but strictly dependent on the cell types. Putative pyramidal neurons appeared more static in FamMap since the early IS period, while putative interneurons were more static only in the late period, during which they switched from moderately dynamic to purely static. Splitting the NovelMap sessions in two phases to distinguish the learning and post-learning periods revealed a rapid transition toward a static scheme of the putative interneurons and an overall increase of stability in the putative pyramidal cells. These results suggest that the prefrontal microcircuitry shifted toward a purely static scheme as soon as the learning was completed, and that such static scheme persisted when the associations became familiar across days, as seen in the FamMap. There is currently a lack of data on the activity of interneurons during learning in monkey studies, with the exception of a study in the inferior temporal cortex[89], it is unclear whether our findings can be generalized to other forms of learning. However, recent studies in the lateral prefrontal cortex have demonstrated a prominent role of putative interneurons to encode information crucial for the flexible learning of new stimulus-reward contingencies in a color-based learning task. In this task, the monkeys, after the presentation of two stimuli, had to identify the movement direction for the stimulus associated with a specific color, which was changed in consecutive reversal blocks. Putative interneurons encoded the stimulus's color to be monitored and modulated their activity during the progression of learning of the new color-reward contingencies after reversals[90,91], suggesting a pronounced recruitment in learning periods, consistent with our evidence. The available studies in mice did not address the problem of coding stability of interneurons; they only show that numerous classes of interneurons play a modulatory role in S-R association tasks in several brain areas involved in associative learning[92–94], and that their activity can change as an effect of learning[93].

Another important aspect to consider is the advantage of a more stable coding scheme following associative learning with familiar S-R. The static coding scheme we reported for putative pyramidal cells can optimize the flow of information to downstream neurons and generate an extended representation in the different microcircuits involved, both temporally invariant[50,87,48,48] and easily extractable[15,48,50] for a downstream neuron. To support such mechanism, interneurons can provide fine control over the probability and timing of pyramidal cell discharges through local synchronization of inhibitory inputs[95] and generate a persistent memory signal by pyramidal neurons during delay periods[68,79,96,97]. Such a function may be generated both locally and at the network level of communication between PF with the caudate nucleus, putamen via corticostriatal loops[98,99], and hippocampus[100], which exhibited to engage bidirectional communication during associative learning. Associative learning also promotes a plastic reconfiguration of the PF microcircuitry with enhanced synchronous connectivity with other areas involved in associative behavior, where cell types specific dopaminergic stimulation is critical[101–105].

Based on our findings, we propose that prolonged experience with familiar associations may generate a plastic adaptation that is cell type specific, which could result in the increased stability of interneurons, possibly providing a temporally extended refinement and sharpening of the pyramidal neurons selectivity after learning. The increase of the pyramidal cells' stability both during and after learning may facilitate the generation of a time-invariant downstream signal, supported by interneurons, easy to extract in all learning phases. Collectively, these mechanisms could improve the PF signal-to-noise ratio in communication with areas involved in associative behavior, leading to rapid and effective recall and maintenance of familiar

associations in memory, ultimately enhancing behavioral performance.

One limitation of the cell types classification approach is that it has been previously shown that it is possible to have interneurons that could be misclassified with hybrid properties, particularly those with waveforms of potentially overlapping intermediate widths between broad and narrow spiking populations and low discharge frequency[46]. Further studies have also identified pyramidal cells called intrinsically bursting[45] and chattering[42] with waveforms reduced in amplitude and high firing, and pyramidal cells of the corticospinal tract with the same confounding characteristics currently identified only in the motor cortex[106]. Considering these past studies, we should be aware that cell classification with our methods can lead to some residual contamination in our analysis, as in other studies in the literature. In spite of these method limitations, other studies have provided support for the reliability of this classification method. Krimer and colleagues[46], investigating the electrophysiological properties and their morphological correlates in the dorsolateral cortex (areas 46 and 9) of 12 macaques, confirmed that most of the cells with high firing (fast-spiking) and reduced waveforms were GABAergic basket and chandelier cells, while cells with regular firing and extended waveforms were mostly pyramidal cells. These studies, together with studies with antidromic stimulation[33] in the PF of the macaque and juxtacellular recording[107], confirm the reliability of classification methods based on the metrics used in the current study.

A second limitation of our study concerns the classification of putative interneurons as a unique class of cells. Recent studies have classified the major BW and NW classes into additional subclasses suitable for identifying subpopulations of separate types of interneurons and pyramidal cells[38,47,54,91]. In rodents, molecular classification of interneuron subtypes according to their specific expression of parvalbumin (PV) and somatostatin (SOM) revealed a major occurrence of such subtypes in the medial prefrontal cortex and a role in working memory[108]. Such subtypes, particularly the PV and partly the SOM interneurons, showed high firing levels and narrow waveforms[108] similar to the properties found in the interneurons' population reported in our study. The challenge for future studies, using those methodologies, requires linking these functional subclasses to morphological and molecular classes of cell types. Such effort might further distinguish the coding schemes and coding magnitude specific to different cell subtypes, and enrich the characterization of the prefrontal microcircuitry's coding properties.

The third limitation was not considering the specific coding properties for stimulus, response, or their combination of the cells selective for the S-R associations, but we know from previous studies that prefrontal cells show associative properties[9]. Asaad and colleagues[9] found that about half of the task-related cells in PF demonstrated non-linear and linear selectivity for arbitrary S-R associations coupled with a minority of cells with specific stimulus and response selectivity during an S-R association reversal task. Although in our study stimulus and response were not dissociated within the mapping tasks, they were studied separately in the non-learning context of the Strategy task. Future studies using a reversal task might overcome this limitation, but only to a degree, considering that association reversal tasks can generate a proactive inhibition of the preceding learned associations on the learning of the reversed associations[9,109] that could affect the cells coding properties.

The fourth limitation, which is common to most studies of the prefrontal cortex, is that when analyzing the response, we could not distinguish neural signals for goals from those for motor plans to achieve those goals, as well as from attentional signals. Indeed, the dissociation of such signals necessitates dedicated experiments[49,110,111].

In conclusion, our results provide new insight into the role of the prefrontal microcircuitry in S-R associations, stimuli and responses, suggesting a key role of the interneurons with their strong and

dynamic coding scheme, particularly in the early phase of the IS period when the integration of the information within the decisional process takes place. We also identified a change in the coding scheme from dynamic to static in this class of cells when the associations become familiar and well established, strictly dependent on the cell types, indicating a flexible adaptation of local interneuron circuitry depending on the learning process. More generally, we found that the presence of mixed dynamics, as described recently by Enel et al.[50] in the PFC, can be dependent on the cell types with the highest stability, observed in the pyramidal neurons, for the representations of stimuli, responses, and their association.

## Methods

### Monkeys and surgery

Two adult male rhesus monkeys (*Macaca mulatta*) of 8.8 and 7.7 kg were used for this experiment. Before the beginning of each experimental session, the monkeys were seated in a primate chair, and the head was fixed with the face stably turned to a screen placed 32 cm from the monkeys' eyes. All procedures conformed to the Guide for the Care and Use of Laboratory Animals (1996) and were approved by the National Institute of Mental Health Animal Care and Use Committee. Both monkeys were anesthetized with isoflurane and underwent a craniotomy on the right frontal lobe. A recording chamber was implanted over the exposed dura mater using several titanium bone screws fixed to the adjacent bone and methacrylate cement together with a head fixation system. Following the surgery, analgesia was given for 3-5 days.

### Experimental tasks

Figure 1a shows the temporal sequence of events common to the three tasks used in this study. Each trial of each task began with the presentation of a fixation spot in the center of the screen (white circle, 0.7° viewing angle) on which the animal had to fix and maintain the fixation (±7.5°) for 1.0 s, together with three spatial targets presented on the right, left and top from the center of the screen (14° from the center of the screen). Subsequently, the fixation spot was removed and replaced by a visual instruction stimulus in the same position that the monkey had to continue fixating on for a time of 1, 1.5, or 2 s selected pseudorandomly. The subsequent disappearance of the IS acted as a go signal for performing a saccade to the selected targets and maintaining fixation for 1 s. Then, all three targets turned white, the monkey was required to maintain fixation for another 0.5 s, and if appropriate, 0.1 ml of reward fluid was released at the end of this period. Finally, the targets turned off in both rewarded and unrewarded trials, and the 2.5 s intertrial period began. Each IS was composed of two American standard codes for information interchange (ASCII) characters of different colors superimposed and generated pseudorandomly.

### Strategy task

The Strategy task was characterized by two classes of trials and two strategies that the monkeys were required to use according to the task rule (Fig. 1b). A repeat trial was a trial in which the same IS from the previous trial was presented in the current trial, requiring the monkey to select the same target as in the previous trial (repeat-stay strategy). A change trial was a trial in which the IS of the previous trial differed from the IS presented in the current trial, requiring the animal to reject the previously selected target and select one of the two alternative targets (change-shift strategy).

A trial was considered correct if the required strategy was successfully implemented. In the change trials, only one of the two remaining targets was randomly associated with a reward. When in a change trial, a target was chosen according to the change-shift strategy, but the target was not rewarded, although it was strategically correct (first chance trial). The same trial was presented again (second

chance trial) until the monkey selected the rewarded target. In the standard version of the Strategy task, the probability of getting a reward was 50% in first chance trials, while in the high-reward version of the Strategy, the probability rose to 90%. The Strategy task analyzed in this study included both of the aforementioned versions.

If the monkey made a strategic error, a correction trial followed where the same trial was represented (as in the not rewarded change-shift trial) until the correct target was chosen. Importantly, each IS could be associated with a different spatial target, consequently preventing any learning of a fixed stimulus-response association by task design.

### Novel and familiar mapping task
Unlike the Strategy task in the novel and familiar mapping tasks, each of the three ISs was associated with a specific fixed target (Fig. 1b). The monkeys were required to map the association between each IS and one of the three targets. The difference between the two tasks lies in the fact that in the NovelMap task, the three ISs were unique and generated at the beginning of each recording session, and the monkeys had to learn three new stimulus-response mappings each time. In contrast, in the FamMap task, three ISs were highly familiar with the same learned stimulus-response mapping presented through the recording sessions. After a correct response, a reward was released in both tasks, and in the case of an error no reward was released. A correction trial followed an incorrect trial in which the same mapping was represented until a correct response was made.

### Data collection methods and histological analysis
A quartz-insulated platinum-iridium electrode (80 μm outer diameter; impedance, 0.5–1.5 MΩ at 1 kHz) was used to isolate single-unit potentials, advanced into the cortex by a 16-electrode microdrive with independent control of each electrode (Thomas Recording, Giessen, Germany) through a custom, concentric recording head with 518 μm electrode spacing. The signal from each electrode was amplified and discriminated using a Multispike Detector (Alpha-Omega Engineering, Nazareth, Israel) or a Multichannel Acquisition Processor (Plexon, Dallas, TX). NIHM CORTEX was used for task presentation, behavioral control, and data collection. The monkeys' eye position was recorded and monitored with an infrared oculometer (Bouis Instruments, Karlsruhe, Germany).

The daily recording sessions were divided into blocks of approximately 100 trials, each consisting of the three tasks used. Usually, the first block started with the standard version of the Strategy task, followed by the NovelMap task and the high reward version of the Strategy task, both using the new set of ISs created. Finally, the Fam-Map task was presented using a set of highly familiar ISs for the animal and was stable between recorded sessions.

At the end of the data collection, two electrolyte lesions (15 μA for 10 s, anodal current) were induced by two penetrations at two different depths. Approximately ten days after this procedure, the animal was anesthetized and perfused with buffered formaldehyde (3% by weight), and steel pins were inserted at the known coordinates of the chamber. The brain was cut coronally into 40 μm sections on a freezing microtome and Nissl-stained for cytoarchitectonic analysis. Steel pins and electrolytic lesions were considered references to trace the surface projections of the recording sites.

### Behavioral analysis for learning trial estimation, and learning and post-learning block definition in NovelMap
To identify the trial in which learning was completed for each set of novel associations in the NovelMap task sessions, we applied a moving average method[53] defined as follows:

$$p_k = (2w+1)^{-1} \sum_{i=k-w}^{k+w} n_i \qquad (1)$$

This method generates for each trial $k$ a window size of $(2w+1)$ trials, where $k$ is the central trial. Such a window is a binary vector, where 1 denotes the $k$ trials when each association was performed correctly, and 0 otherwise. Binomial distribution was employed to identify the window (and learning trial $k$) in which the $p_k$ was significantly higher ($p < 0.05$) than the $p_{null}$, that is the probability of correct under the null hypothesis. We used $w = 2$, which dictated a 5-trial window and a $p_{null} = 0.45$, requiring that all 5 trials be correct to identify the $k$ learning trial. For this analysis, we selected the sessions with at least 42 trials recorded since the start of the session. Finally, for each session, we split the trials into two blocks to apply further analyses: the learning block, which included trials from the first to the $k$ trial identified by the algorithm, and the post-learning block, which included the trial after the $k$ trial to the end of the session.

### Waveform analysis and data preprocessing
The raw signal was sampled at 40 kHz and filtered with a 600–6000 Hz bandpass filter to extract spike activity. A spike threshold method was used to identify the putative single-unit activity, setting the threshold to reduce the possibility of capturing multiunit activity. Single-cell potentials were isolated offline (Off-Line Sorter, Plexon) using different selection criteria: a clear clustering of spike waveforms and isolation in three-dimensional PCA space, lack of interspike intervals <1 ms, and stable maintenance of discharge activity throughout the entire recording session[7]. For this study, waveforms identified in the spike sorting phase from the same isolated single unit recorded through different recording blocks within the same daily recording session were merged into a single sample for the subsequent cell types classification analysis phase. The complete dataset underwent a further manual curation to select only the most isolated single units with a canonical mean waveform (1643/1789). We then recalculated the average waveform of each unit, keeping only the spike waveforms that did not exceed 3 standard deviations from the initial average waveform at each point, in order to increase the accuracy in the subsequent cell types classification procedure and to remove noisy waveforms[54,112]. Finally, to obtain the dataset used for neuronal analyses (1457/1643), we removed the cells with a main trough amplitude smaller than the amplitude of the next peak and with the amplitude of the previous peak 20% greater than the main trough[54] to remove waveforms that might come from axons, typically characterized by an intrinsic short duration that could result in a wrong classification of cell types[113,114]. To classify the cells of our dataset into broad and narrow waveforms spiking cells, we implemented a method used in previous studies[38,90,115] (performed using the waveform analysis toolbox). We applied cubic interpolation to each cell of the final dataset to increase the sampling accuracy of the average waveform from 25 μs to 2.5 μs. The interpolated waveforms were then normalized and aligned to the main trough. We computed the trough-to-peak duration as a cell classification metric, representing the distance between the main trough and the following waveform peak. Two different versions of the Hartigan dip test were used to statistically test the bimodality of the trough-to-peak distribution: the original Hartigan dip test[116] and its calibrated version capable of ensuring greater sensitivity[38]. We calculated two Akaike's and Bayesian information criteria indices to determine whether the trough-to-peak distribution could be fitted by a one- or two-Gaussian model. For the two-Gaussian model, two cutoffs were set, which divided the distribution into three parts. The point within the distribution where the probability of being classified as a narrow cell was at least 10 times greater than the probability of being classified as a broad cell represented the first cutoff. Similarly, the second cutoff was set when the probability of being classified as broad was 10 times greater than a classification as a narrow cell[38]. Cells belonging to the distribution area that fell between the two cutoffs were considered unclassifiable.

## Firing metrics

To characterize the intrinsic firing behavior for each neuron, we calculated 3 different firing metrics from the fixation period (0.0 - 1.0 s) (Fig. 1a) and using all the completed trials: firing rate, Fano factor, and coefficient of variation (CV). The CV is defined as:

$$CV = \frac{\sigma_{ISI}}{\mu_{ISI}} \qquad (2)$$

where $\sigma_{ISI}$ is the standard deviation of interspike intervals and $\mu_{ISI}$ is the mean of the interspike intervals. The Fano factor is defined as:

$$\text{Fano factor} = \frac{\sigma^2_{\text{spike count}}}{\mu_{\text{spike count}}} \qquad (3)$$

where $\sigma^2_{\text{spike count}}$ is the variance of the spike count and $\mu_{\text{spike count}}$ is the mean of the spike count.

The CV and Fano factor are measures of variability that estimate the regularity of the spike train with values close to or less than 1, indicating a trend of regular discharge and values greater than that an irregular discharge pattern[38,47]. The median of the distributions of neurons classified as narrow and broad spiking was then calculated on the values of each metric.

## Population analysis

To quantify the information conveyed by the broad and narrow populations, we used the percentage of explained variance (ω2)[27,51,117](analysis implemented using the measures of effect size (MES) toolbox, version 1.6.0.0), which allowed us to calculate the amount of variance in the neurons' firing rate explained by the tested variable. $\omega^2$ is defined as:

$$\omega^2 = \frac{SS_{\text{Between Groups}} - df * MSE}{SS_{\text{Total}} + MSE} \qquad (4)$$

$SS_{\text{Between Groups}}$ is the sum of squares between groups (variance between groups):

$$SS_{\text{Between Groups}} = \sum_{i=1}^{G} n_i(\bar{x}_i - \bar{x})^2 \qquad (5)$$

where G is the total number of group, and $n_i$ and $\bar{x}_i$ are the number of trials and the average activity of the $i$-th group.

$SS_{\text{Total}}$ is the total sum of squares (total variance):

$$SS_{\text{Total}} = \sum_{i=1}^{N} (x_i - \bar{x})^2 \qquad (6)$$

where $N$ represents the total number of trials for the cell. MSE denotes the mean squared error (variance within groups):

$$MSE = \sum_{i=1}^{G} \sum_{j=1}^{n_i} (x_{i_j} - \bar{x}_i)^2 \qquad (7)$$

df denotes the degrees of freedom (i.e., the levels of the variable of interest – 1). To account for the bias in the calculation of $\omega^2$, for each neuron, we balanced the number of trials in each condition of the analyzed variable to the lowest common value among them, and we used this value to randomly sample the trials for each condition. This procedure was repeated 50 times, and the mean value was calculated between repetitions. We calculated the $\omega^2$ in bins of 150 ms resampled every 15 ms from the start of the analysis window. To investigate the selectivity latencies between cell types populations and associative tasks, we identified the first significant bin from the IS onset for each variable of interest and each cell.

## Cross-temporal decoding analysis

A decoding analysis was implemented using a methodology developed by Meyers et al.[118] (implemented using the neural decoding toolbox, version 1.0.4) and used in previous studies[28,29]. For each neuron, the binary spike activity was averaged in 50 ms bins sampled at 50 ms intervals and the trials labeled for the conditions of the variable. We decided to use a maximum correlation coefficient linear classifier to discriminate between the experimental conditions for its limited computational requirements and its properties that make it resistant to variations in the firing rate of the analyzed populations[29]. Classification accuracy was used as a metric to evaluate the classifier's ability to discriminate the experimental conditions. This metric consists of the number of correctly predicted test trials (during each cross-validation) when the correlation coefficients between tests and training trials belonging to the same condition were higher than when they belong to different conditions, divided by the total number of conditions tested. We applied a 10-fold cross-validation (unless otherwise stated), where for each experimental condition and neuron, 10 trials were randomly selected, of which 9 were used to train the classifier and a single trial was used to test the classifier in predicting the associated condition. The procedure was then repeated 10 times using a different test trial each time, and the prediction results were averaged between repetitions. During the cross-validation procedure, z-score normalization was applied to ensure an equal contribution of the entire neuronal population regardless of the discharge rate by subtracting the mean from each neuron activity and dividing by the SD (both calculated in the training trials considering all conditions) in the training and test trials[29,119]. Because the classifier's performance can be influenced by the size of the population used, we have balanced the broad and narrow spiking populations for each variable analyzed. At this point, the entire procedure was repeated 50 times, each time randomly resampling the trials and neurons, and the results were averaged over the cross-validations and over these 50 runs.

We trained and tested the classifier using all possible combinations of 50 ms time bins to establish the temporal evolution of information coding. This analysis results in a classification accuracy matrix where the values along the diagonal are calculated by performing training and testing on equivalent time bins. In contrast, different time bins are used to calculate the off-diagonal values. For graphic purposes only, we normalized the cross-temporal matrix by rescaling the maximum and minimum accuracy values to a range of values between zero and one.

## Static classification of cross-temporal time points and stability index

To investigate how the variables of interest were coded in time at the broad and narrow spiking population level, we classified the static off-diagonal time points of cross-temporal decoding by implementing the method used in previous studies[26,55,56]. We first calculated the difference in accuracy between each off-diagonal time point, obtained by training and testing the classifier at two different time bins of 50 ms each, and the two on-diagonal time bins corresponding to the time bins used for training and testing. The accuracy value at each on-diagonal time bin was obtained by training and testing the classifier at the same time bin. The same procedure was applied to the 1000 iterations of the null distribution (see Statistical analysis, method 1), allowing us to obtain a distribution of the differences in accuracy values between the off- and on-diagonal bins. Next, each off-diagonal time point was classified as static if such differences were lower than 99.9% of the differences estimated at the corresponding time point in the null distribution (cluster-based permutation test, $p < 0.001$), and the accuracy value was significantly higher than chance level (cluster-based permutation test, $p < 0.001$).

The formula below formalizes our definition of a static time bin:

$$\text{Static binary matrix}_{(tp1,tp2)} = \sim \left( \text{CTD}_{(tp1,tp2)} < \text{CTD}_{(tp1,tp1)} \right) \wedge$$
$$\sim \left( \text{CTD}_{(tp1,tp2)} < \text{CTD}_{(tp2,tp2)} \right) \wedge \text{CTD}_{(tp1,tp2)} > \text{CTD shuffled labels}_{(tp1,tp2,:)}$$

(8)

where CTD represents the accuracy matrix of cross-temporal decoding obtained on the correctly labeled data, CTD shuffled labels represent the three-dimensional accuracy matrix obtained by randomly shuffling the labels (where the third dimension includes the 1000 iterations performed), tp represents the indices within the matrix (as well as the time bins used to train and test the classifier), and $\vee$ denotes the logical operator AND. Moreover, to classify the off-diagonal time points as static, the two corresponding time bins on-diagonal used for training and testing the classifier were both required to be significantly above chance level (permutation test, $p < 0.0025$, Bonferroni corrected for the number of on-diagonal time bins). Finally, we obtained a binary matrix of the same size as the cross-temporal decoding, where we assigned 1 to the time bins resulting as static and 0 for all remaining time bins.

We then quantified the magnitude of the population coding stability by the stability index. The stability index provides information about the proportion of off-diagonal time points that were classified as static. For each on-diagonal time bin, we calculated the stability index by averaging the two dimensions of the binary matrix, where along the corresponding row and column the classifier was trained for a specific time bin and tested for the remaining bins. We then applied a smoothing procedure to the stability index calculation; that is, we used a moving average in which we included the previous and the next time bin for a given on-diagonal time bin. This procedure allowed us to also use the average of the previous and next row and column within the binary matrix for a given time bin. Moreover, the on-diagonal time bins were required to be significantly above the chance level. Conversely, the stability index was not calculated because it could not provide information on the representation of the variable of interest. Following this, we considered only the time bins significant for both populations to obtain a correct comparison between the stability indices calculated for the broad and narrowspiking populations.

An index of 1 indicates that the variable in the specific time bin was represented in a full static way. An index progressively lower than 1 indicates a progressively lower static representation. The stability index was calculated for both narrow and broad populations separately. To quantify the variability of the stability index, we implemented a bootstrapping method[56]. We repeated the decoding analysis 50 times by resampling with replacement of the neurons of each broad and narrow spiking population (where, for each, we calculated the null distribution by randomly shuffling the experimental conditions 1000 times). We repeated the procedure for calculating the stability index described above, obtaining 50 indices for each population. Finally, we calculated the standard deviation of these 50 indices.

### Statistical analysis
Unless otherwise indicated, the statistical analyses were performed using a cluster-based non-parametric permutation test, a statistical method that checks the statistical significance by taking into account the multiple comparisons performed over time[51,55,120]. To do this, a null distribution was calculated in different ways according to the analysis: 1. randomly shuffling the experimental conditions before performing the decoding analysis and $\omega^2$, repeating this procedure 1000 times to test the significance of each time point concerning chance; 2. randomly shuffling broad and narrow spiking neurons to test the difference in the coding magnitude in the $\omega^2$ analysis; 3. due to the expensive computational demands required by the decoding analysis, we

randomly mixed the broad and narrow spiking neurons 46 times (where for each iteration we shuffled the labels of the experimental conditions 1000 times), and calculated the stability index for each false population obtained. Then, we calculated the 1035 possible differences, considering all the combinations for each time point of the index, in order to compare the stability values of the two correctly classified populations. Contiguous time points in the observed populations that exceeded the 99.9 percentile of the null distribution were considered to be candidate clusters (unless otherwise stated). The maximum summed statistical cluster test[120] was calculated for the observed data and compared with that obtained from the values obtained from the null distribution, and the number of values of the null distribution was greater than the value obtained with the observed data determined the $p$ value of the test. We accepted (unless otherwise stated) as significant only clusters of time points that exceeded the 99.9 percentile of the null distribution (which is equivalent to a $p$ value < 0.001).

### Reporting summary
Further information on research design is available in the Nature Portfolio Reporting Summary linked to this article.

### Data availability
The data necessary for the evaluation of this study are provided with the data source file and the additional data provided in the https://osf.io/bwnq9/ repository. Raw data are available on request from the corresponding author. Source data are provided with this paper.

### Code availability
Cell types were classified using The waveform analysis toolbox (https://bitbucket.org/sardid/waveformanalysis/src/master/). Analysis of explained variance was calculated using The measures of effect size (MES) Toolbox (https://github.com/hhentschke/measures-of-effect-size-toolbox) and decoding analysis with the neural decoding toolbox (http://www.readout.info/). Custom code used to classify static data points and calculate stability index is available at https://osf.io/bwnq9/. Data were analyzed using MATLAB 2021b.

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

## Acknowledgements

We thank Mr. Alex Cummings for the histological material, Dr. Andrew Mitz for engineering, and P. Brasted and Dr. Steve Wise for their numerous contributions. This work has been partially supported by the Sapienza University of Rome (Progetto H2020: PH1181642DB714F6 to A.G. and PH120172B9427FA1 to E.B.), Avvio alla Ricerca 2022 (AR22218167E30967 to L.F.).

## Author contributions

A.G. performed research; F.C. and A.G. designed research; F.C. data curation; F.C. and A.G. writing—original draft; F.C., L.F., F.L., S.R., E.B. analyzed data; F.C., A.G., L.F., F.L., S.R., E.B. writing—review and editing.

## Competing interests

The authors declare no competing interests.
