## [Peer Review File · Nature Communications]

REVIEWER COMMENTS

Reviewer #1 (Remarks to the Author):

In the manuscript titled “Mixed prefrontal cortex dynamic during associative learning: contribution of putative pyramidal cells and interneurons” by Ceccarelli and colleagues, the authors recorded the activity of the dorsolateral prefrontal neurons across 3 tasks. They found that putative pyramidal cells have, on average, less information about task variables, but more stable codes, compared to putative interneurons. Further, they showed that familiarity with one of the tasks leads to some changes in these properties.

Overall I found that the manuscript was clearly written and very interesting. In particular the different stability profiles of putative pyramidal and interneurons. The results/interpretations of the changes associated with learning were weaker but still interesting.

I don't have any major comments. Just minor ones, outlined below:

P4: “The analysis of the amount of variability explained by both the population of BR and NR cells revealed that both groups of neurons significantly encoded the S-R association in both the NovelMap (cluster-based permutation test, $p < 0.001$) and FamMap (cluster-based permutation test, $p < 0.001$) tasks throughout the duration of the IS period (Fig. 3A-B, blue and red bars, for BR and NR populations, respectively).”

I am not sure I agree that this analysis can be interpreted as a metric of encoding the S-R association. Rather, this could represent information about the stimulus, the upcoming response plan, or the association. I would suggest renaming this to better reflect the information.

P4: “Interestingly, the coding in the two tasks differed for latency and peak of activity between the two populations. In the NR population, the associations were coded with shorter population latency and a clear peak in the early phase of the IS period than in the BR population.”

There are no statistics (as far as I can tell) to support this claim.

P5: "We then repeated the ω^2 analysis, confirming the results shown in Fig. 3. For each variable, the coding was significant in the whole IS period, as it was the greater coding of the NR population"

Could you show these plots in the supplementary figures? Also, I don't understand what this means: "(Strategy response, cluster-based permutation test, $p < 0.001$; first bin value: 145 ms, last bin value: 715 ms; Strategy stimulus, $p < 0.001$, 115 ms, 880 ms; NovelMap, $p < 0.001$, 160 ms, 580 ms; FamMap, $p < 0.001$, 130 ms, 385 ms)."

P5: "To test this hypothesis"

Which hypothesis?

P5: "the absence of an off-diagonal reduction in the prediction capacity of the classifier by comparing it with the prediction obtained when the training and test bins were identical, meaning in the on-diagonal values"

You don't really quantify the absence, since that is just a possible result of the quantification. I'd rephrase this sentence to accurately reflect what you quantified (i.e. we quantified the difference between on- and off-diagonal prediction accuracy)

P13: "For graphic purposes only, we normalized the cross-temporal matrix by rescaling the maximum and minimum accuracy values to a range of values between zero and one."

Why was this done? It would be useful to see the actual accuracy values to evaluate how predictive these classifiers actually are (see next comment).

P5: For figs 4-6, the stability metric is a bit hard to interpret without knowing what are the actual (non-normalized) values of the decoder. The stability metric is dependent on the decoding accuracy since non-significant decoding bins are not included in the stability calculation. So simply showing these stability values does not tell us if the differences in stability are related to differences in the information encoded by the populations, or whether it is independent of that and simply reflective of stability by itself.

P5: "The NR populations showed a significantly lower and reduced stability (Fig. 4E), suggesting that a moderate local and dynamic coding scheme was maintained for the whole IS epoch."

Does lower and reduce mean different things? If not, please remove one of the descriptions. Likewise for local and dynamic.

Reviewer #2 (Remarks to the Author):

The manuscript "Mixed prefrontal cortex dynamic during associative learning: contribution of putative pyramidal cells and interneurons" by Ceccarelli et al. analyzes the encoding of Stimulus-Response rules and strategy associations with sensory cues in dorsolateral and dorsomedial prefrontal cortices in two rhesus monkeys using extracellular recordings. The study finds that narrow spiking neurons encode SR rules stronger and earlier as evident in more explained variance early after onset of an instructional cue than broad spiking neurons (Fig 3). The study then reports with a linear classifier that narrow spiking neurons have less temporal stability in encoding SRs rule and cue-related strategy information compared to broad spiking neurons as evident in a more temporally specific decoding accuracy (Fig 4+5).

The study applies important control analysis including a firing rate matching analysis for the encoding of SR rules and controlling for the number of cells contributing to the decoding analysis.

Overall, the findings of the special coding strength and latencies of narrow spiking neurons is important and interesting.

The impact of study is reduced by not explicitly considering the learning period of task.

The study emphasizes the population level of the analysis without describing the distribution of encoding across recorded cells. This approach risks of including neurons in the analysis that are not apparently task relevant or task modulated which may underlie the relatively low average explained variance, which would need to be tested/validated (please see below).

These and a few other aspects of the study deserve consideration to enhance the impact and generalizability of the findings. The aspects are listed below.

- The current manuscript is risking to infer results from a population level response that does not hold when looking at specific groups of neurons that are particularly modulated by the task. The studies' results indicate that broad spiking neurons overall encode less information about SR-mapping or strategy, but that the decidability is more stable over time. It is not clear why this finding is important assuming that the average population response of the neurons did not encode strongly the SR rules. Can this be discussed?

What seems missing is a result that convinces a reader that subgroups of neurons encode the SR rules sufficiently strongly. Such a best-set analysis may identify broad (and narrow) spiking neuron subsets that have strong and dynamic encoding. The conclusion/interpretation of the paper would be different.

E.g. it might be that broad spiking neurons are more heterogeneous in their coding of the SR or strategy information.

- It is not apparent why the learning period of the task is not analyzed separately. This is unclear particularly because the title of the manuscript insinuates that results pertain to "during associative learning" but the correlation with learning is not analyzed separately (and behavioral and neural learning curves are not shown).

- The introduction mentions only in one sentence that cell types may have different contributions by writing "contribution of cell types to associative learning and how the plastic adaptation induced by extended learning affects microcircuitry are still an open question 47." But since the question of cell type specific coding is the central aspect of the paper this is ideally be introduced and described more explicitly to motivate the study. It would help e.g. to already discuss in the introduction some of the papers referred to in the discussion that previous studies report narrow spiking neurons having higher coding and learning correlates while others could not find a difference.

- It is not clear why the response period of the task is not analyzed to confirm that narrow spiking neurons have more transient responses. Are neurons encoding the SR rule in response to the cue not also maintaining their information until the response can be elicited? Analysis or discussion of this aspect would be appreciated to see the bigger picture on how neurons support performance of the task.

- The section "Population results" describes explained variance for "S-R associations". It is unclear what this variable reflects. What were the variables used to predict firing rate here: Was it the difference of correct and error trials, or were the three S-R associations coded separately so that explained variance is the information about which S-R was linked to the instruction stimulus? Were incorrect trials included or not? Were potentially confounding variables like the spatial location of the response target considered to exclude that laterality biases are confounded with SR rule learning?

- The results describe some important findings in general terms without providing supporting statistics. E.g. The authors state "In the NR population, the associations were coded with shorter population latency" but no test - statistics is described and no statistical test applied to support the supposed faster latency (and it is not apparent visually). Such statements either will need a direct statistical test or a more careful wording.

- The explained variances in the encoding analysis seem rather low (below 1% of variance explained). Why so the authors think is so little variance explained compared to what others report? Can a

histogram or scatterplot of the explained variances across cells be shown (e.g. for 300 ms after versus the 300 ms before the IS ? THIS would allow discerning the underlying distribution.

- The study recorded in "dorsolateral and dorsomedial" prefrontal cortex, but does not report whether the coding for the novmapping, fammapping or strategy task varies with anatomical recording site. Considering the potential anatomical specificity seems important given that various previous studies show functional specificity of prefrontal subareas.

- Throughout the paper, neuronal results are only shown in terms of explained variance and calculated indices. Please show some actual firing rate changes relative to the IS of neurons whose activity may reflect the main effects. This is important to enhance the confidence that the reported findings are reflected in single neuron contributions instead of "only" reflecting some averaging effect.

- What were the proportions of neurons that showed individually significant explained variance for the SR or strategy rules?

- One inclusion criterion for neurons was that they were isolated for >10 trials. Was there an exclusion criterion on the minimum firing rate required for the analysis. Would including only stronger firing neurons (e.g. >2Hz max firing during any epoch in the task) change the results?

- The learning performance is briefly summarized in the results section, but can the learning curves be shown directly to discern how fast animals reached above chance performance? E.g. by adding the learning behavioral curves to Fig. 1.

- the abstract is written not very concisely. The last sentence in the abstract is rather vague and does not give the reader a hint on why the cell type differences in coding would matter.

- Why are figure 1 panels B and C/D differently sized?

- In the literature (eg. Mitchel 2007 Neuron) narrow spiking cells have typically been labeled in red (not blue). Can this be adjusted (Fig. 1).

- Please spell out what " IS " means in the text before abbreviating (p 3)

- There are two recent studies on narrow spiking neurons in the lateral prefrontal cortex that seem to relate closely to and support the current findings and approach and might want to be considered. Oemisch et al., 2019 Nat Commun. reported narrow spiking neurons code stronger for learning related prediction errors and carry stimulus specific information, and Banaie Boroujeni et al., 2021 eLife report that narrow spiking neurons have a suppressive influence on multiunit activity and correlate with learning in response to a visual cue.

Reviewer #3 (Remarks to the Author):

The authors investigate the coding properties of functionally distinct neuronal populations recorded in the prefrontal cortex during an associative task.

The recorded single units were clustered based on waveform shape and firing statistics in putative excitatory (BR) and inhibitory neurons (NR).

They presented either familiar or novel stimulus-response associations and reported that the putative interneurons encoded the associations in both task types more strongly and earlier. Furthermore, they employed a strategy task that dissociated stimulus and response components and found that the finding generalized to this non-associative task.

Additionally, the authors examined the temporal evolution of neural coding stability and identified that BR population had a more stable representation of the SR associations than the NR population.

Overall the results support the conclusions made by the authors.

Excitatory cells tend to have a higher selectivity to the stimulus features (e.g color or shape of the S) than narrow-waveform neurons. I wonder whether the result shown in Figure 3 (BR neurons explained variability is lower than NR) can be explained by the fact that these neurons have higher stimulus selectivity, hence presenting a higher trial-by-trial response heterogeneity. If that is the case, the explained variance may be high in trials where the stimulus features match the neurons' tuning and lower in the trials where the other stimuli are presented. On the other hand, narrow-waveform neurons, if having a lower preference for the stimulus features, may encode with similar strength the different stimuli.

To test this, the authors could show the same plots as in Figure 3, but compute the explained variance separately for each stimulus presented.

The authors show that the narrow-waveform cells coding scheme increases in stability in the FamMap compared to the NovelMap (Fig. 6). It would be interesting to characterize the time dynamics of this shift further. For instance, after how many FamMap sessions the NR cells shift from a dynamic to a stable coding, and after how many repetitions does this change in coding stability reaches a plateau?

Also, do the authors see any increase in the coding stability in the NR population along a single session of NovelMap task?

Both interneurons and excitatory cells are heterogeneous classes with distinct functional properties. Those specializations are evident, for instance, in their different firing statistics. It would be interesting if the authors show the coding properties of individual neurons and not only the average to identify if there are sub-clusters having different coding schemes. For instance, different interneuronal types (e.g. Somatostatin interneurons) have broad-waveforms but may present different coding properties than excitatory cells.

The authors should explicitly explain how the explained variance is computed in the “Strategy Response” (Fig. 3C). As far as I understood, the authors calculated the amount of variability explained by the fact it is a repeat-stay or change trial, but it seems not clearly stated in the paper.

Could the author provide information regarding the depth of electrode insertion?

It is unclear why the acronyms NR and BR were chosen for the narrow and broad spiking cells. I suggest using the more commonly used acronym NW and BW.

In plots A and B from Fig.2, it seems that the latency in the coding of the associations decreases in the FamTask compared to the NovelMap task in the NR population. It would be interesting to mention it and quantify such latency difference.

We would like to thank the reviewers and the editor for the positive evaluation of our work. The constructive comments have led to a significant improvement in the manuscript. We have provided point-by-point responses to the reviewers' comments. We have included two additional analyses and 8 supplementary figures (10 in total now), plus 7 rebuttal figures specific for reviewers, along with modifications to the main figures. We introduced a new methods section to describe the identification of the learning and post-learning period in the NovelMap task. We also added an analysis for statistical comparison of coding latency for all variables of interest and between learning tasks, as requested by all the reviewers. In response to some reviewers' comments, we have extended the section on the limitations of our study, now renamed "Interpretational issues and limitations of the study". In accordance with some of the reviewers' comments and to align with the cell-types literature informal conventions, we have replaced the previous acronyms BR and NR, designating broad and narrow spiking cells with BW and NW, respectively. This change has been consistently applied in our responses to the reviewers to prevent any potential confusion. Likewise, we changed the color code associated with each cell type: previously blue was associated with narrow spiking cells and red with broad spiking cells. Now after the inversion of the colors, blue represents broad spiking cells and red narrow spiking cells in all the figures in the article. The references added in the manuscript were included in the bibliography. Additionally, references exclusively used in this document to address reviewers' comments have been included at the end of the file.

Finally, we thank the reviewers for finding a mistake in plotting the percentages of explained variance on the y-axis of Fig. 3, accidentally reported as a proportion (maximum value 1) and not as a percentage. We have corrected current Fig. 3 and reported the percentages of explained variance in all current figures.

Reviewer #1 (Remarks to the Author):

In the manuscript titled "Mixed prefrontal cortex dynamic during associative learning: contribution of putative pyramidal cells and interneurons" by Ceccarelli and colleagues, the authors recorded the activity of the dorsolateral prefrontal neurons across 3 tasks. They found that putative pyramidal cells have, on average, less information about task variables, but more stable codes, compared to putative interneurons. Further, they showed that familiarity with one of the tasks leads to some changes in these properties.

Overall I found that the manuscript was clearly written and very interesting. In particular the different stability profiles of putative pyramidal and interneurons. The results/interpretations of the changes associated with learning were weaker but still interesting. I don't have any major comments. Just minor ones, outlined below:

We thank the reviewer for the overall positive feedback on our manuscript. Point-by-point responses to the comments follow below.

I don't have any major comments. Just minor ones, outlined below:

P4: "The analysis of the amount of variability explained by both the population of BR and NR cells revealed that both groups of neurons significantly encoded the S-R association in both the NovelMap (cluster-based permutation test, $p < 0.001$) and FamMap (cluster-based permutation test, $p < 0.001$) tasks throughout the duration of the IS period (Fig. 3A-B, blue and red bars, for BR and NR populations, respectively)."

I am not sure I agree that this analysis can be interpreted as a metric of encoding the S-R association. Rather, this could represent information about the stimulus, the upcoming response plan, or the association. I would suggest renaming this to better reflect the

information.

We agree with the reviewer's point that the mapping tasks did not allow to dissociate stimuli, responses, and specific combinations of stimuli and responses, and we have acknowledged this limitation in the now re-titled "*Interpretational issues and limitations of the study*" section of the discussion. On the other hand, the Strategy task allowed the dissociation of stimulus and response, although not in the learning context. However, some insight comes from the article of Asaad et al. (1998), which we now cite, showing that about half of the prefrontal neurons are selective for the combination of stimuli and responses, not just for the stimulus or the response. We need to consider that dissociating the two variables remains challenging and unnatural even when using a learning reversal task, as done by Asaad et al. (1998). In fact, even reversal tasks pose interpretation problems, as Cromer et al. (2011) pointed out: "Reversing stimuli means that, in addition to learning new cue-response associations, monkeys must inhibit the old learned associations that are no longer correct. Therefore, proactive interference exists between the former associations (no longer relevant) and the new associations."

We did not find an alternative short terminology to propose for indicating that the neural coding in the learning tasks can reflect the coding of different aspects of the S-R association, but we made a further effort to explain what we intend for coding of the S-R associations. Accordingly, we added the following text in the results: "*For novel and familiar S-R associations in the NovelMap and FamMap tasks, we analyzed only correct trials where each of the three stimuli was associated with only one response following the S-R mappings defined by the specific task rule. Associations between stimuli and responses were not changed during the mapping sessions. Consequently, we did not distinguish the contribution of response and stimulus cell selectivity to the coding of the S-R associations. However, as shown later, we used the Strategy task to study the coding of different cell types for stimulus and response independently.*". And reported this limitation in the newly expanded section titled "*Interpretational issues and limitations of the study*" in the discussion as follows: "*The third limitation was not considering the specific coding properties for stimulus, response, or their combination of the cells selective for the S-R associations, but we know from previous studies that prefrontal cells show associative properties⁹. Asaad and colleagues⁹ found that about half of the task-related cells in PF demonstrated non-linear and linear selectivity for arbitrary S-R associations coupled with a minority of cells with specific stimulus and response selectivity during an S-R association reversal task. Although in our study stimulus and response were not dissociated within the mapping tasks, they were studied separately in the non-learning context of the Strategy task. Future studies using a reversal task might overcome this limitation, but only to a degree, considering that association reversal tasks can generate a proactive inhibition of the preceding learned associations on the learning of the reversed associations^{9,108} that could affect the cells coding properties.*".

P4: "Interestingly, the coding in the two tasks differed for latency and peak of activity between the two populations. In the NR population, the associations were coded with shorter population latency and a clear peak in the early phase of the IS period than in the BR population."

There are no statistics (as far as I can tell) to support this claim.

We appreciate the reviewer's keen observation regarding the necessity for statistical tests to assess latency differences. In response, we have introduced a new analysis to evaluate the selectivity onset of the individual neurons in the two cell type populations, and we statistically compared the two distributions of latencies.

We have added the following paragraph to describe the analysis performed in the methods: "*To investigate the selectivity latencies between cell types populations and associative tasks, we identified the first significant bin from the IS onset for each variable of interest and each cell.*".

We performed a latency analysis by comparing the latency of the selectivity between the NW and BW populations for the variables studied and between NovelMap and FamMap. The results confirmed a significant earlier involvement of the NW population in coding the task variables in the mapping tasks. However, we observed just a non-significant tendency in the Strategy task. Interestingly, we also identified a cell type-specific effect for the NW population when comparing the coding latencies between the two mapping tasks. We reported these results in the results section as follows: *“To test for a difference in the selectivity latencies, we identified the first significant time bin within the IS period in the two cell type populations. We found that NW population coded significantly earlier, both novel (NovelMap, mean onset time, from IS onset: NW 233.0 ms; BW 281.1 ms, Kruskal-Wallis between time onset distributions, $p < 0.05$) and familiar associations (FamMap, mean onset time, from IS onset: NW 179.1 ms; BW 265.4 ms, Kruskal-Wallis between time onset distributions, $p < 0.05$). Moreover, we found a significant decrease in latencies of familiar versus new associations coding, in the NW population (Kruskal-Wallis between NW time onset tasks distributions, from IS onset, $p < 0.05$), but not in the BW population (Kruskal-Wallis between BW time onset tasks distributions, from IS onset, $p > 0.05$).”*.
“Latency analyses revealed a tendency in the NW population to encode both the response (Strategy response, mean onset time, from IS onset: NW 289.3 ms; BW 327.1 ms, Kruskal-Wallis between time onset distributions, $p > 0.05$) and the stimulus (Strategy stimulus, mean onset time, from IS onset: NW 229.2 ms; BW 283.5 ms, Kruskal-Wallis between time onset distributions, $p > 0.05$) earlier in the Strategy task, when compared to the BW population, however without achieving statistical significance.”.

P5: “We then repeated the ω^2 analysis, confirming the results shown in Fig. 3. For each variable, the coding was significant in the whole IS period, as it was the greater coding of the NR population”

Could you show these plots in the supplementary figures? Also, I don't understand what this means: “(Strategy response, cluster-based permutation test, $p < 0.001$; first bin value: 145 ms, last bin value: 715 ms; Strategy stimulus, $p < 0.001$, 115 ms, 880 ms; NovelMap, $p < 0.001$, 160 ms, 580 ms; FamMap, $p < 0.001$, 130 ms, 385 ms).”

We thank the reviewer for pointing this out. We intended to report the first and last significant time bins where the coding was higher for the narrow population than the broad population for each variable. We have modified the text to make it clearer as follows: *“(Strategy response, cluster-based permutation test, $p < 0.001$; first bin time: 145 ms, last bin time: 715 ms; Strategy stimulus, $p < 0.001$, 115 ms, 880 ms; NovelMap, $p < 0.001$, 160 ms, 580 ms; FamMap, $p < 0.001$, 130 ms, 385 ms, from IS onset) (Supplementary Fig. 3a, d, g, j, black bars).”*. We have also added the plots of this analysis in the Supplementary Fig. 3, along with the figure already present with the results of the cross-temporal decoding.

Supplementary Figure 3. Cross-temporal decoding, stable data point classification, stability index, and explained variance in populations with matched firing. Novel associations in NovelMap (a, b, c), familiar associations in FamMap (d, e, f), and response (g, h, i) and stimulus (j, k, l) in the Strategy task. The organization of the figure is the same as that in Fig. 3 and Fig. 4 for explained variance and cross-temporal decoding analysis, respectively. Times are shown in milliseconds.

P5: “To test this hypothesis”

Which hypothesis?

We have replaced this phrase with the text as referred to in our response to the next point.

P5: “the absence of an off-diagonal reduction in the prediction capacity of the classifier by comparing it with the prediction obtained when the training and test bins were identical, meaning in the on-diagonal values”

You don’t really quantify the absence, since that is just a possible result of the quantification. I’d rephrase this sentence to accurately reflect what you quantified (i.e. we quantified the difference between on- and off-diagonal prediction accuracy)

Thank you for the suggestion, we have now rephrased the sentence as follows: “We quantified and statistically tested the difference between the on- and off-diagonal prediction accuracies of the classifier by comparing the off-diagonal prediction accuracies with the predictions obtained when the training and test bins were identical, that are the on-diagonal values (see Methods).”.

P13: “For graphic purposes only, we normalized the cross-temporal matrix by rescaling the

maximum and minimum accuracy values to a range of values between zero and one.” Why was this done? It would be useful to see the actual accuracy values to evaluate how predictive these classifiers actually are (see next comment).

P5: For figs 4-6, the stability metric is a bit hard to interpret without knowing what are the actual (non-normalized) values of the decoder. The stability metric is dependent on the decoding accuracy since non-significant decoding bins are not included in the stability calculation. So simply showing these stability values does not tell us if the differences in stability are related to differences in the information encoded by the populations, or whether it is independent of that and simply reflective of stability by itself.

We chose to use this graphical representation format for two reasons. The first, as evident in the figure added in this response with the non-normalized values, is that the accuracy levels vary between variables of interest and cell types. Consequently, it is challenging to adopt a unique and effective color scale to visualize the on- and off-diagonal differences that our analysis captures for all variables of interest. The second reason, closely related to the first, stems from our normalization approach that considers the maximum and minimum values of the maps to make the relative differences more visible within each map between on- and off-diagonal (which has the maximum accuracy values) accuracies that as mentioned it is the key feature underlying the calculation of the stability index, which considers such relative differences rather than absolute differences in accuracy levels.

However, as the reviewer proposed, we have added an additional supplementary figure in the revised manuscript with the original non-normalized accuracy values of the maps in Fig 4 and 5.

We added the following text in the results when referring to the figure: “In addition, Supplementary Fig. 6 shows non-normalized cross-temporal decoding matrices”.

Supplementary Figure 6. Cross-temporal decoding non-normalized classification accuracy. Color maps corresponding to the analysis in Fig. 4, showing non-normalized classification accuracy values for (a) new associations in NovelMap, (b) familiar associations in FamMap, and (c) response, and (d) stimulus in the Strategy task. Times are shown in milliseconds.

P5: "The NR populations showed a significantly lower and reduced stability (Fig. 4E), suggesting that a moderate local and dynamic coding scheme was maintained for the whole IS epoch."

Does lower and reduce mean different things? If not, please remove one of the descriptions. Likewise for local and dynamic.

We thank the reviewer for indicating this potential source of confusion. We actually used "lower" and "reduced" together to reinforce the concept of lower stability of the narrow population, and both were intended to convey the same meaning, likewise "local" and "dynamic." To avoid the confusion, we changed the phrase keeping only "lower" and "dynamic": "*The NW population showed significantly lower stability (Fig. 4E), suggesting that a moderately dynamic coding scheme was maintained for the whole IS period.*".

Reviewer #2 (Remarks to the Author):

The manuscript "Mixed prefrontal cortex dynamic during associative learning: contribution of putative pyramidal cells and interneurons" by Ceccarelli et al. analyzes the encoding of Stimulus-Response rules and strategy associations with sensory cues in dorsolateral and dorsomedial prefrontal cortices in two rhesus monkeys using extracellular recordings. The study finds that narrow spiking neurons encode SR rules stronger and earlier as evident in more explained variance early after onset of an instructional cue than broad spiking neurons (Fig 3). The study then reports with a linear classifier that narrow spiking neurons have less temporal stability in encoding SRs rule and cue-related strategy information compared to broad spiking neurons as evident in a more temporally specific decoding accuracy (Fig 4+5).

The study applies important control analysis including a firing rate matching analysis for the encoding of SR rules and controlling for the number of cells contributing to the decoding analysis.

Overall, the findings of the special coding strength and latencies of narrow spiking neurons is important and interesting.

We thank the reviewer for the acknowledging the importance and providing a positive assessment of our work and the control analyses performed. We also appreciate the constructive comments that allowed us to enrich the manuscript with further controls to confirm the main findings and interpretations. Point-by-point responses to the comments follow below.

The impact of study is reduced by not explicitly considering the learning period of task.

We thank the reviewer for this comment. We believe our new analysis strengthens the manuscript to identify the learning period and the comparison with the following post-learning period within the NovelMap sessions. Introducing these new results enriched the characterization of cell types' selectivity and stability during and after learning and also revealed the timing of the change in the coding scheme of the interneurons.

The study emphasizes the population level of the analysis without describing the

distribution of encoding across recorded cells. This approach risks of including neurons in the analysis that are not apparently task relevant or task modulated which may underlie the relatively low average explained variance, which would need to be tested/validated (please see below).

To test whether the effect could be accounted for by non task-related cells, we performed the main analyses (Fig. 3 and Fig. 4) only on task-related cells. We have added this control analysis to the manuscript. Overall, our results generalized to these restricted populations of cells. The following text is now added to the results section: “We next assessed whether the presence of cells not modulated by the tasks reduced the observed ω^2 values. We defined neurons as task-related if their firing rate in the baseline period (fixation period: 0.0-1.0 s) differed significantly (Kruskal-Wallis, $p < 0.05$) from any of the two phases (early IS period: 50-400 ms; late IS period: 400-900 ms) of the IS period (Strategy response, BW 567/664 and NW 200/226; Strategy stimulus, BW 574/674 and NW 203/232; NovelMap, BW 377/487 and NW 151/190; FamMap, BW 201/275 and NW 84/98). We found that overall the BW and NW task-related populations had ω^2 values and a higher coding magnitude in the NW population, comparable to the results in Fig. 3 (Supplementary Fig. 4a, d, g, j).”. We have also added the following text in the results section: “Analogously, task-related BW and NW populations and populations with a firing rate greater than 0.5 Hz maintained a coding scheme consistent with those found in the main results (Supplementary Fig. 4, 5 and 9c-d, respectively)”.

Supplementary Figure 4. **Cross-temporal decoding, stable data point classification, stability index and explained variance for task-related cells.** Novel associations in NovelMap (a, b, c), familiar associations in FamMap (d, e, f), and response (g, h, i) and stimulus (j, k, l) in the Strategy task. The

organization of the figure is the same as that in Fig. 3 and Fig. 4 for explained variance and cross-temporal decoding analysis, respectively. Times are shown in milliseconds.

These and a few other aspects of the study deserve consideration to enhance the impact and generalizability of the findings. The aspects are listed below.

- The current manuscript is risking to infer results from a population level response that does not hold when looking at specific groups of neurons that are particularly modulated by the task. The studies' results indicate that broad spiking neurons overall encode less information about SR-mapping or strategy, but that the decidability is more stable over time. It is not clear why this finding is important assuming that the average population response of the neurons did not encode strongly the SR rules. Can this be discussed? What seems missing is a result that convinces a reader that subgroups of neurons encode the SR rules sufficiently strongly. Such a best-set analysis may identify broad (and narrow) spiking neuron subsets that have strong and dynamic encoding. The conclusion/interpretation of the paper would be different. E.g. it might be that broad spiking neurons are more heterogeneous in their coding of the SR or strategy information.

We thank the reviewer for bringing this aspect to our attention and also to the Fig. 3 axes where we accidentally reported the proportion (maximum values 1 and not 100 as a percentage) and not the percentage of explained variance. We have corrected all the current figures. As you can see now, broad spiking neurons explain on average, from 2 to 3 % of the explained variance. These values are in line with those reported for similar task-related information in studies that however did not divide cell types in the lateral and dorsolateral prefrontal cortex (Wasmuht et al., 2018; Kadohisa et al., 2023; Buschman et al., 2011). We reported this link to previous works in the discussion as follows: "*Overall, the percentage of variance explained by the task variables is consistent with that reported in previous studies in PFC^{51,66,67}*".

We agree with the reviewer that this could be a concern. As we report in the following response in more detail, the higher selectivity is in line with the significantly higher percentage of NW neurons, which are selective for the variables compared to BW neurons. This stronger engagement of the NW population underlies the higher coding observed in Figure 3, which now includes these observations. We next calculated the explained variance for the restricted populations of selective cells with a significant effect for at least one of the two epochs where selectivity was tested (Early: 50-450 ms and late: 500-900 ms IS period, Cluster-based permutation test, $p < 0.05$). The figure below (Rebuttal Figure 1-2) shows that such populations were highly selective for all the variables. We can see that the stability indices and cross-temporal decoding maps do not differ from those in Fig. 4 and 5. Together, these results indicate that cell type populations maintain a homogeneous temporal coding scheme regardless of the magnitude of coding expressed by the populations tested.

What is changing instead is the coding magnitude for the two classes of cells in the NovelMap task and for the stimulus in the Strategy task. This indicates that the observed higher population coding in these cases, when considering the entire population, was primarily influenced by a larger number of significant cells within the NW population, as shown in the updated Figure 3. In the FamMap task and in the response coding in the Strategy task instead, we observed higher coding of the significantly NW selective cells in addition to a greater percentage of NW significant cells as shown in the updated Figure 3.

Rebuttal Figure 1. Cross-temporal decoding, stable data point classification, stability index, and explained variance for selective cells. Novel associations in NovelMap (a, b, c), familiar associations in FamMap (d, e, f), and response (g, h, i) and stimulus (j, k, l) in the Strategy task. The organization of the figure is the same as that in Fig. 3 and Fig. 4 for explained variance and cross-temporal decoding analysis, respectively. Times are shown in milliseconds.

Rebuttal Figure 2. Cell-types stability index comparison between associative tasks for selective cells. New associations in NovelMap (dashed lines) and familiar associations in the FamMap task (solid lines) are directly compared for the BW (a) and NW (b) populations, respectively. Black bars indicate a significant difference in the stability levels between the two tasks (cluster-based permutation test < 0.001). Times are shown in milliseconds.

- It is not apparent why the learning period of the task is not analyzed separately. This is unclear particularly because the title of the manuscript insinuates that results pertain to "during associative learning" but the correlation with learning is not analyzed separately (and behavioral and neural learning curves are not shown).

We thank the reviewer for suggesting such analysis. We initially disregarded the possibility of performing such an analysis as we were concerned about not having enough trials to perform it, but we found out that it was still possible, at least on a subpopulation of cells. We agree that a further analysis of the initial learning period of the novel associations in the NovelMap task, combined with a comparison of the subsequent period after learning is completed and the associations become more familiar, can strengthen the main findings. For such analysis, we needed to identify the trial where learning was accomplished in each session. To do this, we used a commonly adopted method to estimate that trial. We included the following text in the methods to describe the method in detail:

"Behavioral analysis for learning trial estimation and learning and post-learning block definition in NovelMap

To identify the trial in which learning was completed for each set of new associations in the NovelMap task sessions, we applied a moving average method⁵³ defined as follows:

$$p_k = (2w + 1)^{-1} \sum_{i=k-w}^{k+w} n_i$$

This method generates for each trial k a window size of $(2w + 1)$ trials, where k is the central trial. Such a window is a binary vector, where 1 denotes the k trials when each association was performed correctly, and 0 otherwise. Binomial distribution was employed to identify the window (and learning trial k) in which the p_k was significantly higher ($p < 0.05$) than the p_{null} , that is the probability of correct under the null hypothesis. We used $w=2$, which dictated a 5-trial window and a $p_{null} = 0.45$, requiring that all 5 trials be correct to identify the k learning trial. For this analysis, we selected the sessions with at least 42 trials recorded since the start of the session. Finally, for each session, we split the trials into two blocks to apply further analyses: the learning block, which included trials from the first to the trial preceding the k trial identified by the algorithm, and the post-learning block, which included the k trial to the end of the session.”.

Correspondingly, we have edited the behavioral results section to include and report the results of this analysis as follows: “We applied a moving average method⁵³ (see Methods) to identify the trial when learning was completed in the NovelMap task. We found that both monkeys quickly learned the three S-R associations (Fig. 1d), on average within 16 ± 0.7 trials (monkey 1: 17.3 ± 1.1 trials, monkey 2: 14.3 ± 0.9 trials, Wilcoxon rank-sum test between monkeys $p > 0.05$), with an average overall performance of 93.0% correct responses over all trials (monkey 1: 93.6%, monkey 2: 92.4%).”. **We have added in Fig. 1 the learning curves of the NovelMap, including a reference to the trial in which, on average, the learning was completed, calculated by the algorithm. We have incorporated the analysis in the results as follows:** “To further study the cell types' contribution in the associative coding during the learning and the post-learning periods, where associations are consolidated, we identified in each NovelMap session the trial when learning was completed and accordingly split the recorded sessions into two blocks of trials: the learning (from the first trial to the learning completion trial) and the post-learning block (from the trial after the learning completion trial to the last one recorded in the session). In this analysis, we selected the cells from the previously analyzed sample with at least 6 trials for each association in the learning (NovelMap learning block, NW: 42/210; BW: 87/580) and post-learning blocks (NovelMap post-learning block, NW: 152/210; BW: 376/580). The results observed mirrored the selectivity profiles found between NovelMap and FamMap using the whole session (Supplementary Fig. 2a-d). Populations of both the cell types encoded the associations over the entire IS period in both the learning and post-learning blocks (cluster-based permutation test, $p < 0.001$). In the learning block, the NW population encoded the associations stronger than the BW population for most of the IS period (cluster-based permutation test, $p < 0.001$; first bin time: 220 ms, last bin time: 775 ms, from IS onset). However, in the post-learning block, the stronger selectivity, was restricted mostly to the early IS period (cluster-based permutation test, $p < 0.001$; first bin time: 70 ms, last bin time: 595 ms, from IS onset), to vanish later.”.

“We then repeated the analysis using the learning and post-learning blocks of trials within the NovelMap sessions. Interestingly, in the learning block (Supplementary Fig. 2b-c) the NW population showed a significant reduction of the levels of stability (cluster-based permutation test, $p < 0.001$), suggesting the use of a dynamic coding scheme during the IS period, compared to the high stability levels achieved in the late IS period by the BW population, indicating a purely static scheme. However, in the post-learning block (Supplementary Fig. 2e-f), we found an overall increase in stability comparable to that reported in FamMap, where the BW population showed a significant rise in stability in the early phase of the IS period compared to the learning block (Supplementary Fig. 8a). The NW population in the post-learning block, although with lower levels of stability than the BW population (cluster-based permutation test, $p < 0.001$) (Supplementary Fig. 2e-f) exhibited a strong increment in stability compared to the previous learning block (Supplementary Fig. 8b). Such changes in cell types coding schemes are consistent with those found between the entire NovelMap and FamMap sessions, again suggesting a shift from a dynamic to a

static coding scheme in the NW population even before the associations become familiar. These results taken together suggest that such a coding scheme shift may occur fast after learning for novel mappings to be maintained thereafter.”.

Finally, we have added a part to the discussion on the implications of the results of this analysis:

“Splitting the NovelMap sessions in two phases to distinguish the learning and post-learning periods revealed a rapid transition toward a static scheme of the putative interneurons and an overall increase of stability in the putative pyramidal cells. These results suggest that the prefrontal microcircuitry shifted toward a purely static scheme already after the learning was completed, which persisted when the associations became familiar across days, as seen in the FamMap”.

Supplementary Figure 2. Cross-temporal decoding, stable data point classification, stability index and explained variance in within-session NovelMap. Learning block of trials (a, b, c), and post-learning block of trials (d, e, f) in the NovelMap task. The organization of the figure is the same as that in Fig. 3 and Fig. 4 for explained variance and cross-temporal decoding analysis, respectively. We applied number-matching procedure (see Methods), by selecting 40 cells for each cell type population and trials block. In the classification maps each data point (on-diagonal points are not considered in this analysis and left in blue by convention) are in yellow if static or blue if are not (see Methods). (c, f) Coding stability over time is quantified by the stability index for each cell type, where high values correspond to pronounced stability (see Methods). Shaded areas represent 1 bootstrapped standard deviation of the index (see Methods). Black bars show a significant difference in stability between the two populations (cluster-based permutation test < 0.001). Times are shown in milliseconds.

Supplementary Figure 8. Cell-types stability index comparison between associative tasks in within-session NovelMap. Learning block of trials (dashed lines) and post-learning block of trials in the NovelMap task (solid lines) are directly compared for the BW (a) and NW (b) populations, respectively. Black bars indicate a significant difference in the stability levels between the two tasks (cluster-based permutation test < 0.001). Times are shown in milliseconds.

- The introduction mentions only in one sentence that cell types may have different contributions by writing "contribution of cell types to associative learning and how the plastic adaptation induced by extended learning affects microcircuitry are still an open question 47." But since the question of cell type specific coding is the central aspect of the paper this is ideally be introduced and described more explicitly to motivate the study. It would help e.g. to already discuss in the introduction some of the papers referred to in the discussion that previous studies report narrow spiking neurons having higher coding and learning correlates while others could not find a difference.

We agree with the reviewer that highlighting the background that motivated our work may improve the introduction. Therefore, we have added this part in the introduction: "Similarly, the contribution of cell types in driving the magnitude of associative coding in PFC microcircuitry

remains uncertain, since conflicting evidence has been presented while studying different task-related information in PFC and other cortical areas^{31-34,36,47}".

- It is not clear why the response period of the task is not analyzed to confirm that narrow spiking neurons have more transient responses. Are neurons encoding the SR rule in response to the cue not also maintaining their information until the response can be elicited? Analysis or discussion of this aspect would be appreciated to see the bigger picture on how neurons support performance of the task.

[Redacted]

[Redacted]

- The section "Population results" describes explained variance for "S-R associations". It is unclear what this variable reflects. What were the variables used to predict firing rate here: Was it the difference of correct and error trials, or were the three S-R associations coded separately so that explained variance is the information about which S-R was linked to the instruction stimulus? Were incorrect trials included or not? Were potentially confounding variables like the spatial location of the response target considered to exclude that laterality biases are confounded with SR rule learning?

We thank the reviewer for mentioning this missing information. We used only correctly performed trials for all the analyses reported in the manuscript. In NovelMap and FamMap, we selected trials in which the stimulus presented and the response were uniquely associated with one of the three possible associations. We added a description of trial selection for each variable in the results section as follows: *"For the response and the stimulus in the Strategy task, we selected the correct trials in which the response corresponded to one of the three possible correct target positions (top, right, left) and the three stimuli displayed, respectively. For novel and familiar S-R associations in the NovelMap and FamMap tasks, we analyzed only correct trials where each of the three stimuli was associated with only one response following the S-R mappings defined by the specific task rule. Associations between stimuli and responses were not changed during the mapping sessions. Consequently, we did not distinguish the contribution of response and stimulus cell selectivity to the coding of the S-R associations. However, as shown later, we used the Strategy task to study the coding of different cell types for stimulus and response independently."*

We did not dissociate stimuli, responses, and associations in our NovelMap and FamMap tasks, and we acknowledged it in the now re-titled "Interpretational issues and limitations of the study" section of the discussion. However, we learned from the article by Asaad et al. (1998) that about half of the neurons show selectivity for the combination of stimuli and response, not just a simple stimulus or response selectivity. To answer the question on the laterality bias a preference for associations for contralateral responses would be shared between learning tasks and between the different phases of the learning and should not represent a confound for the interpretation of the results.

Also, in response to another reviewer's question we added the following text in the newly expanded section titled "Interpretational issues and limitations of the study" in the discussion: *"The third limitation was not considering the specific coding properties for stimulus, response, or their combination of the cells selective for the S-R associations, but we know from previous studies that prefrontal cells show associative properties⁹. Asaad and colleagues⁹ found that about half of the task-related cells in PF demonstrated non-linear and linear selectivity for arbitrary S-R associations coupled with a minority of cells with specific stimulus and response selectivity during an S-R association reversal task. Although in our study stimulus and response were not dissociated*

within the mapping tasks, they were studied separately in the non-learning context of the Strategy task. Future studies using a reversal task might overcome this limitation, but only to a degree, considering that association reversal tasks can generate a proactive inhibition of the preceding learned associations on the learning of the reversed associations^{9,108} that could affect the cells coding properties.”.

- The results describe some important findings in general terms without providing supporting statistics. E.g. The authors state "In the NR population, the associations were coded with shorter population latency " but no test - statistics is described and no statistical test applied to support the supposed faster latency (and it is not apparent visually). Such statements either will need a direct statistical test or a more careful wording.

We thank the reviewer for pointing out this weakness of the article. We implemented a new analysis where we identified the selectivity time onset for each cell and compared the two onsets distributions obtained for the NW and the BW populations. We added a paragraph in the methods as follows: “To investigate the selectivity latencies between cell types populations and associative tasks, we identified the first significant bin from the IS onset for each variable of interest and each cell.”. We then added the following text in the results: “To test for a difference in the selectivity latencies, we identified the first significant time bin within the IS period in the two cell type populations. We found that NW population coded significantly earlier, both novel (NovelMap, mean onset time, from IS onset: NW 233.0 ms; BW 281.1 ms, Kruskal-Wallis between time onset distributions, $p < 0.05$) and familiar associations (FamMap, mean onset time, from IS onset: NW 179.1 ms; BW 265.4 ms, Kruskal-Wallis between time onset distributions, $p < 0.05$). Moreover, we found a significant decrease in latencies of familiar versus new associations coding, in the NW population (Kruskal-Wallis between NW time onset tasks distributions, from IS onset, $p < 0.05$), but not in the BW population (Kruskal-Wallis between BW time onset tasks distributions, from IS onset, $p > 0.05$).”.

“Latency analyses revealed a tendency in the NW population to encode both the response (Strategy response, mean onset time, from IS onset: NW 289.3 ms; BW 327.1 ms, Kruskal-Wallis between time onset distributions, $p > 0.05$) and the stimulus (Strategy stimulus, mean onset time, from IS onset: NW 229.2 ms; BW 283.5 ms, Kruskal-Wallis between time onset distributions, $p > 0.05$) earlier in the Strategy task, when compared to the BW population, however without achieving statistical significance.”.

- The explained variances in the encoding analysis seem rather low (below 1% of variance explained). Why so the authors think is so little variance explained compared to what others report? Can a histogram or scatterplot of the explained variances across cells be shown (e.g. for 300 ms after versus the 300 ms before the IS ? THIS would allow discerning the underlying distribution.

As we have explained before, by mistake, we did not correctly report the percentage of explained variance in the axes of Fig 3. In the new version of the manuscript, we have corrected this mistake. As shown now in Fig 3, the percentage of explained variance for the BW population exceeds 1%. To enrich the cell types' encoding characterization we added a scatterplot for each variable as a Supplementary Fig. 1, showing the relations between the explained variance of each neuron in the 500 ms prior to IS onset (late fixation period) and the subsequent first 500 ms of the IS period (early IS period). The results show, as expected, that for both cell types, selective cells exhibited a strong increase in explained variance during the IS early period, especially when compared with the previous baseline fixation period that represents an epoch of the tasks in which variables cannot yet be represented. In reference to the Supplementary Fig. 1, we added the following text in the results section: “Explained variance cells' distributions in the investigated sample for novel

and familiar associations confirmed an important involvement of both cell type populations in the IS early period and the highest population coding of the NW population (Supplementary Fig. 1a-b). "The distribution of the explained variance confirms the involvement of both cell types along with a higher selectivity of the NW population (Supplementary Fig. 1c-d)".

Supplementary Figure 1. Explained variance of cell-type populations between the late fixation and the early IS period. Scatterplots display the explained variance values for each cell of the last 500 ms of the fixation period and the following 500 ms of the early IS period, for NovelMap (a), FamMap (b), and for response (c) and stimulus (d) in the Strategy task. Distributions in each plot represent the explained variance values in the fixation (top) and IS periods (right). Asterisks represent a significant difference between the cell encoding distributions of cell types, indicating a higher coding magnitude in the NW population. Kruskal-Wallis test: **: $p < 0.01$; ***: $p < 0.001$.

- The study recorded in "dorsolateral and dorsomedial" prefrontal cortex, but does not report whether the coding for the novmapping, fammapping or strategy task varies with anatomical recording site. Considering the potential anatomical specificity seems important given that various previous studies show functional specificity of prefrontal subareas. In response to this comment, we divided the recording sessions for each task by the location of their penetrations in the dorsolateral prefrontal cortex (PFdl, Brodman area 46) and the dorsomedial cortex (PFdm, area 9). Plus, we selected for the analysis only the cells with at least 10 trials for each condition of the variables of interest (Strategy response, PFdl: BW 433, NW 92, PFdm: BW 169, NW 89; Strategy stimulus, PFdl: BW 442, NW 96, PFdm: BW 169, NW 90; NovelMap, PFdl: BW 323, NW 84, PFdm: BW 122, NW 71; FamMap, PFdl: BW 186, NW 37, PFdm: BW 58, NW 36). We recalculated the explained variance for each area and we found that overall, as in the main results, both cell types encoded all variables of interest during the IS period. For both areas (as shown in the following Rebuttal Figures 4-5), we found evidence that the NW population coded more for all the variables of interest than BW without finding noticeable differences between the areas. We observed an overall lower magnitude of selectivity, which was discontinuous over time and globally fragmented in statistical significance during the IS period. We believe that was related to the considerable reduction of the neurons sample. Because of the lack of complete, non-discontinuous statistical significance during the IS period, we could not perform

the cross-temporal decoding analysis to quantify the coding scheme of the two cell types populations, given that our definition of temporal coding scheme requires that the variable is coded continuously for at least most of the epoch. For this reason, we decided not to include these results in the manuscript.

Rebuttal figure 4. Explained variance of PFdl cell types for novel and familiar mapping, stimulus, and response. Percentage of variance explained (ω^2) averaged on the BW (blue) and NW (red) populations for the coding of the mapping in the NovelMap (a), FamMap (b), responses (c) and stimuli (d) in the Strategy task. Data aligned to the IS onset, covering 1 s of the IS period and the last 500 ms of the fixation period. Colored dashed lines show ± 1 SEM of ω^2 . Colored bars below the figures mark ω^2 values significantly higher than the null distribution (cluster-based permutation test $p < 0.001$, see Methods); black bars highlight a significant difference between the two populations (cluster-based permutation test $p < 0.001$, see Methods).

Rebuttal figure 5. Explained variance of PFdm cell types for novel and familiar mapping, stimulus, and response. Percentage of variance explained (ω^2) averaged on the BW (blue) and NW (red) populations for the coding of the mapping in the NovelMap (a), FamMap (b), responses (c), and stimuli (d) in the Strategy task. Data aligned to the IS onset, covering 1 s of the IS period and the last 500 ms of the fixation period. Colored dashed lines show ± 1 SEM of ω^2 . Colored bars below the figures mark ω^2 values significantly higher than the null distribution (cluster-based permutation test $p < 0.001$, see Methods); black bars highlight a significant difference between the two populations (cluster-based permutation test $p < 0.001$, see Methods).

- Throughout the paper, neuronal results are only shown in terms of explained variance and calculated indices. Please show some actual firing rate changes relative to the IS of neurons whose activity may reflect the main effects. This is important to enhance the confidence that the reported findings are reflected in single neuron contributions instead of “only” reflecting some averaging effect.

In the Supplementary Fig 7, we have now added an example raster for each studied variable and task that, with their firing patterns, may contribute to the coding schemes identified for the cell types. We have referred to the figure in the main text as follows: “Static and dynamic coding schemes are thought to be the product of an interplay between neurons of the decoded population⁵⁰. Single unit selectivity properties such as duration of selectivity^{28,50,55} and preference switching⁵⁵ contribute to such coding schemes. Supplementary Fig. 7 shows examples of single units classified as BW in NovelMap (Supplementary Fig. 7a) and FamMap (Supplementary Fig. 7b). Each of these cells and the NW cell in FamMap (Supplementary Fig. 7b) exhibited sustained selectivity throughout the IS period associated with a stable coding scheme. Supplementary Fig. 7a shows a NW cell recorded in NovelMap, characterized by a dynamic coding scheme.”.

“Supplementary Fig. 7 shows two BW example cells, selective for response (Supplementary Fig. 7c) and stimulus (Supplementary Fig. 7d) in the Strategy task, whose persistent selectivity indicates a

static coding scheme. Finally, the two NW example cells, with the first exhibiting a transient response selectivity (Supplementary Fig. 7c) and the second a stimulus preference switching (Supplementary Fig. 7d), indicate both a dynamic coding scheme.”.

Supplementary Figure 7. Raster plot cell-types: Single unit activity during late fixation and IS period for BW and NW example cells for new associations in NovelMap (a), familiar associations in FamMap (b), response (c) and stimulus (d) in the Strategy task. The black triangle shows alignment at the IS onset. Colored lines (top part of each raster) show spike density activity for each variable condition.

- What were the proportions of neurons that showed individually significant explained variance for the SR or strategy rules?

As anticipated in the previous reply, we recomputed the explained variance by dividing the IS period in two phases. We considered the cells as selective if they encoded the variables of interest in at least one of the two phases. We added histograms with the percentages of significant cells for each variable in Fig 3. We added the following text in the results: “To further investigate the engagement of BW and NW populations in associative coding, we recomputed ω^2 in two bins reflecting the two phases of the IS period and identified the cells modulated significantly (early: 50-450 ms and late: 500-900 ms IS period, Cluster-based permutation test, $p < 0.05$). As shown by the histograms in Fig. 3a and b, we found an overall larger engagement of the NW population than the BW population, measured in terms of percentages of cells modulated by the S-R associations (NovelMap early IS period, NW: 32%; BW: 16%; NovelMap late IS period, NW: 31%; BW: 17%; FamMap early IS period, NW: 32%; BW: 14%; FamMap late IS period, NW: 21%; BW: 19%).”.

“We found, as for the associative coding, a significantly greater percentage of cells encoding response and stimulus (Fig. 3c and 3d, histograms) in the NW than in the BW population (Strategy response early IS period, NW: 22%; BW: 14%; Strategy response late IS period, NW: 30%; BW: 20%; Strategy stimulus early IS period, NW: 22%; BW: 9%; Strategy stimulus late IS period, NW: 21%; BW: 10%).”.

Fig 3. Cell types explained variance and percentages of selective cells for novel and familiar associations, stimulus and response. Percentage of variance explained (ω^2) averaged on the BW (blue) and NW (red) populations for the associations coding in the NovelMap (a), FamMap (b), responses (c) and stimuli (d) in the Strategy task. Data aligned to the IS onset, covering 1 s of the IS period and the last 500 ms of the fixation period. Colored dashed lines show ± 1 SEM of ω^2 . Colored bars below the figures mark ω^2 values significantly higher than the null distribution (cluster-based permutation test $p < 0.001$, see Methods); black bars highlight a significant difference between the two populations (cluster-based permutation test $p < 0.001$, see Methods). Histograms within each figure display the percentages of significant cells for BW and NW populations in the early (50-450 ms) and late (500-900 ms) phase of the IS period. Chi-square test: **: $p < 0.01$; ***: $p < 0.001$; NS: not significant.

- One inclusion criterion for neurons was that they were isolated for >10 trials. Was there an exclusion criterion on the minimum firing rate required for the analysis. Would including only stronger firing neurons (e.g. >2Hz max firing during any epoch in the task) change the results?

The neurons included in the analyses were required to be recorded for at least 10 trials for each condition of the variable of interest (stimulus, response, novel and familiar associations), where each variable has 3 specific experimental conditions (e.g., stimulus A, B and C). It follows that each cell was recorded for at least 30 trials. We modified the text where we described this selection to explain it. We also thank the reviewer for suggesting such additional control analysis, as we reported in Fig 2b, the BW population has, in line with the literature, a reduced firing rate compared to NW population, in our dataset it is around 1 Hz during the baseline fixation period. For this reason, we could not impose a very restrictive selection criteria on firing rate, and using even a 2 Hz criterion would lead us, on average, to retain only 26% of cells for all variables studied. To reach a compromise between having a minimum statistical power for the analysis and the aim of removing the contribution of low-firing cells, we chose a 0.5 Hz cut-off calculated throughout

the whole session. Overall, the results did not differ from the main results, and we reported this analysis in the results as follows: “Finally, we tested whether the results were affected by cells with low firing rate by removing those cells with a very low firing rate. For this analysis, we eliminated the cells with a firing rate less than or equal to 0.5 Hz throughout the recording session (Strategy response, BW 475/664 and NW 192/226; Strategy stimulus, BW 481/674 and NW 195/232; NovelMap, BW 350/487 and NW 160/190; FamMap, BW 203/275 and NW 81/98). Such selection criterion also did not result in any change in the previously observed effects (Supplementary Fig. 5a, d, g, j).”. “Analogously, task-related BW and NW populations and populations with a firing rate greater than 0.5 Hz maintained a coding scheme consistent with those found in the main results (Supplementary Fig. 4, 5 and 9c-d, respectively)”.

Supplementary Figure 5. Cross-temporal decoding, stable data point classification, stability index and explained variance in cell-types populations with firing greater than 0.5 Hz. Novel associations in NovelMap (a, b, c), familiar associations in FamMap (d, e, f), and response (g, h, i) and stimulus (j, k, l) in the Strategy task. The organization of the figure is the same as that in Fig. 3 and Fig. 4 for explained variance and cross-temporal decoding analysis, respectively. Times are shown in milliseconds.

- The learning performance is briefly summarized in the results section, but can the learning curves be shown directly to discern how fast animals reached above chance performance? E.g. by adding the learning behavioral curves to Fig. 1.

Considering this suggestion, we have added in Fig. 1 the learning curves divided by monkey in

NovelMap and FamMap tasks, respectively. For NovelMap, regarding the learning period analysis described earlier, we have also reported the trial, on average, identified as indicative of learning being completed.

Figure 1. Experimental tasks, recording locations and learning curves. (a) The sequence of events in common between the three experimental tasks. Black squares represent the video screen, the white dot indicates the fixation spot, the empty white squares show the response targets, and the dotted lines represent the gaze of the monkey on the chosen target. (b) Examples of instructor stimulus sets used in the recording sessions in FamMap unchanged throughout the recording days, NovelMap, and Strategy tasks with the stimulus sets changed in each recording session and shared between the two tasks. A sequence of trials is shown in the Strategy task, and the two classes of trials and strategies (repeat and change trial, stay and switch strategy) of the task. Red arrows indicate the correct response for that trial. (c) Penetration sites of the recordings in the two monkeys. (d,e) Percentage of correct choices in the initial 50 trials of the NovelMap (d) and

FamMap (e) task sessions divided by monkey. Horizontal dashed lines indicate the 95 % confidence limits. The vertical solid and dashed lines correspond to the mean trials of learning completion (see Methods) and ± 1 SEM respectively. AS: Arcuate sulcus; PS: Principal sulcus; ASS: Stimulus-response association.

- the abstract is written not very concisely. The last sentence in the abstract is rather vague and does not give the reader a hint on why the cell type differences in coding would matter.

We thank the reviewer for the suggestion. We changed the abstract as follows by emphasizing, briefly, for the reason of the maximum required abstract length, the implications of our findings:

“The prefrontal cortex maintains information in memory through static or dynamic population codes depending on task demands, but whether the population coding schemes used are learning-dependent and differ between cell types is currently unknown. We investigated the population coding properties and temporal stability of macaque neurons recorded in two mapping tasks during and after stimulus-response associative learning, and then used a Strategy task with the same stimuli and responses as control. We identified a heterogeneous population coding for stimuli, responses, and novel associations: static for putative pyramidal cells and dynamic for putative interneurons that showed the strongest selectivity for all the variables. The population coding of learned associations showed overall the highest stability driven by cell types, with interneurons changing from dynamic to static coding after successful learning. The results support that prefrontal microcircuitry expresses mixed population coding governed by cell types and adapts interneurons' stability during associative learning.”.

- Why are figure 1 panels B and C/D differently sized?

We resized Fig 1 b to be the same size as Fig 1 c and d .

- In the literature (eg. Mitchel 2007 Neuron) narrow spiking cells have typically been labeled in red (not blue). Can this be adjusted (Fig. 1).

We reversed the color codes for the cell types in all figures (main and supplementary) so that the manuscript has narrow spiking cells in red and broad spiking cells in blue consistent with the labelling conventions used in the previous articles.

- Please spell out what " IS " means in the text before abbreviating (p 3)

We thank the reviewer for pointing out this. In the introduction, we had already defined IS as an acronym for "instruction stimulus". However, starting with the results, such an acronym was also used for the epoch of analysis, in this case, the acronym was followed by "period" (occasionally denoted as "IS phase", to maintain consistency, we removed this alternative in the manuscript). In order to clarify this distinction for the reader, we defined the epoch of analysis at the beginning of the results section as follows: *“We considered the IS period as the epoch of interest for all subsequent analyses, which includes the period from the IS onset to 1 second after it (see Fig. 1a for the task timeline).”.*

- There are two recent studies on narrow spiking neurons in the lateral prefrontal cortex that seem to relate closely to and support the current findings and approach and might want to be considered. Oemisch et al., 2019 Nat Commun. reported narrow spiking neurons code stronger for learning related prediction errors and carry stimulus specific information, and Banaie Boroujeni et al., 2021 eLife report that narrow spiking neurons

have a suppressive influence on multiunit activity and correlate with learning in response to a visual cue.

We thank the reviewer for bringing these two important papers to our attention. We believe they should be included to our discussion, particularly in the section where we report studies investigating the interneurons' contribution to learning, which is limited in the macaques' studies to the best of our knowledge. We have now added the following paragraph in the discussion on the findings related to the role of different cell types in learning: "However, recent studies in the lateral prefrontal cortex have demonstrated a prominent role of putative interneurons to encode information crucial for the flexible learning of new stimulus-reward contingencies in a color-based learning task. In this task, the monkeys, after the presentation of two stimuli, had to identify the movement direction for the stimulus associated with a specific color, which was changed in consecutive reversal blocks. Putative interneurons encoded the stimulus's color to be monitored and modulated their activity during the progression of learning of the new color-reward contingencies after reversals^{90,91}."

Reviewer #3 (Remarks to the Author):

The authors investigate the coding properties of functionally distinct neuronal populations recorded in the prefrontal cortex during an associative task.

The recorded single units were clustered based on waveform shape and firing statistics in putative excitatory (BR) and inhibitory neurons (NR).

They presented either familiar or novel stimulus-response associations and reported that the putative interneurons encoded the associations in both task types more strongly and earlier. Furthermore, they employed a strategy task that dissociated stimulus and response components and found that the finding generalized to this non-associative task. Additionally, the authors examined the temporal evolution of neural coding stability and identified that BR population had a more stable representation of the SR associations than the NR population.

Overall the results support the conclusions made by the authors.

We thank the reviewer for the positive evaluation of our work. Point-by-point responses to the comments follow below.

Excitatory cells tend to have a higher selectivity to the stimulus features (e.g color or shape of the S) than narrow-waveform neurons. I wonder whether the result shown in Figure 3 (BR neurons explained variability is lower than NR) can be explained by the fact that these neurons have higher stimulus selectivity, hence presenting a higher trial-by-trial response heterogeneity. If that is the case, the explained variance may be high in trials where the stimulus features match the neurons' tuning and lower in the trials where the other stimuli are presented. On the other hand, narrow-waveform neurons, if having a lower preference for the stimulus features, may encode with similar strength the different stimuli.

To test this, the authors could show the same plots as in Figure 3, but compute the explained variance separately for each stimulus presented.

We think there could be a misunderstanding regarding how the explained variance was calculated as a cell selectivity metric for the stimulus variable. As we reported in the methods, the formula is as follows.

$$\omega^2 = \frac{SS_{Between\ Groups} - df * MSE}{SS_{Total} + MSE}$$

The critical part of determining the strength of selectivity involves the calculation of $SS_{Between\ Groups}$, i.e., the sum of the differences between the mean activity of each condition (stimulus A, B, C) and the mean activity considering all trials; which is then counterbalanced by the variability calculated within the groups (i.e. MSE in the formula). The exact same formula was used to estimate selectivity for various task-related information in the prefrontal cortex (Wasmuht et al., 2018; Kadohisa et al., 2023; Buschman et al., 2011). Furthermore, previous studies have shown specific selectivity for complex stimuli in the prefrontal cortex (Asaad et al., 2000; Miller et al., 1996; Asaad et al., 1998). Taking this into account, it is not possible to calculate the explained variance independently for each stimulus. In the case of a neuron with high selectivity, $SS_{Between\ Groups}$ will be larger than that obtained for a nonselective neuron with similar activity between stimulus conditions. We have extended the section dedicated to the explained variance in methods, explaining the formula in more detail as follows: “To quantify the information conveyed by the broad and narrow populations, we used the percentage of variance explained (ω^2)^{27,50,108}, which allowed us to calculate the amount of variance in the frequency of neuronal discharge explained by the tested variable. ω^2 is defined as:

$$\omega^2 = \frac{SS_{Between\ Groups} - df * MSE}{SS_{Total} + MSE}$$

$SS_{Between\ Groups}$ is the sum of squares between groups (variance between groups):

$$SS_{Between\ Groups} = \sum_{i=1}^g n_i (\bar{x}_i - \bar{x})^2$$

Where n_i and \bar{x}_i are the number of trials and the average activity of the i th group.

SS_{Total} is the total sum of squares (total variance):

$$SS_{Total} = \sum_{i=1}^N (x_i - \bar{x})^2$$

Where N represents the total number of trials for the cell. MSE denotes the mean squared error (variance within groups):

$$MSE = \sum_{i=1}^g \sum_{j=1}^{n_i} (x_{ij} - \bar{x}_i)^2$$

df denotes the degrees of freedom (i.e., the levels of the variable of interest – 1). To account for the bias in the calculation of ω^2 , for each neuron, we balanced the number of trials in each condition of the analyzed variable to the lowest common value among them, and we used this value to randomly sample the trials for each condition. This procedure was repeated 50 times, and the mean value was calculated between repetitions. We calculated the ω^2 in bins of 150 ms resampled every 15 ms from the start of the analysis window.”

The authors show that the narrow-waveform cells coding scheme increases in stability in the FamMap compared to the NovelMap (Fig. 6). It would be interesting to characterize the time dynamics of this shift further. For instance, after how many FamMap sessions the NR cells shift from a dynamic to a stable coding, and after how many repetitions does this change in coding stability reaches a plateau?

Also, do the authors see any increase in the coding stability in the NR population along a single session of NovelMap task?

We thank the reviewer for the interesting questions. We agree that it would be important to characterize the timing once learning is consolidated to move from dynamic to static coding. By task design, the monkeys were already overtrained in the training phase, with the same ISs also used in the subsequent FamMap recording phase, as a result starting from the first recording session with FamMap, the ISs already had a strong familiarity. However, to address the reviewer's question we further confirmed the presence of static coding in FamMap throughout the recording period related to the consistently high familiarity of ISs, also given the impossibility of testing the stability session by session due to the small number of neurons recorded simultaneously (on average 5.9 per session), we divided the sessions into two blocks: the first 30 and the last 30 sessions. The stability levels observed in the two blocks of sessions were comparable with each other and with the main results as shown by the figure in the analysis below:

Rebuttal figure 6. Cross-temporal decoding, stable data point classification, stability index for FamMap sessions blocks: Familiar associations in the first 30 and (a, b) last 30 (c, d) sessions recorded in FamMap. The organization of the figure is the same as that in 4. Times are shown in milliseconds.

We also included an additional and very informative analysis that allowed us to split individual NovelMap sessions into blocks of trials where learning is in progress and blocks where learning has concluded. With this method, we were able to find evidence of a shift from a dynamic to a static coding scheme also within individual learning sessions. These new results, although limited to a small sample of neurons, suggest that the temporal dynamics of this transition may be relatively fast during learning. Below, the Supplementary Fig. 2 and 8 where the results of this analysis are shown.

Supplementary Figure 2. Cross-temporal decoding, stable data point classification, stability index and explained variance in within-session NovelMap. Learning block of trials (a, b, c), and post-learning block of trials (d, e, f) in the NovelMap task. The organization of the figure is the same as that in Fig. 3 and Fig. 4 for explained variance and cross-temporal decoding analysis, respectively. We applied number-matching procedure (see Methods), by selecting 40 cells for each cell type population and trials block. In the classification maps each data point (on-diagonal points are not considered in this analysis and left in blue by convention) are in yellow if static or blue if are not (see Methods). (c, f) Coding stability over time is quantified by the stability index for each cell type, where high values correspond to pronounced stability (see Methods). Shaded areas represent 1 bootstrapped standard deviation of the index (see Methods). Black bars show a significant difference in stability between the two populations (cluster-based permutation test < 0.001). Times are shown in milliseconds.

Supplementary Figure 8. Cell-types stability index comparison between associative tasks in within-session NovelMap. Learning block of trials (dashed lines) and post-learning block of trials in the NovelMap task (solid lines) are directly compared for the BW (a) and NW (b) populations, respectively. Black bars indicate a significant difference in the stability levels between the two tasks (cluster-based permutation test < 0.001). Times are shown in milliseconds.

Both interneurons and excitatory cells are heterogeneous classes with distinct functional properties. Those specializations are evident, for instance, in their different firing statistics. It would be interesting if the authors show the coding properties of individual neurons and not only the average to identify if there are sub-clusters having different coding schemes. For instance, different interneuronal types (e.g Somatostatin interneurons) have broad-waveforms but may present different coding properties than excitatory cells.

We thank the reviewer for the constructive comment. We acknowledge that recent studies have offered a further subclassification of the narrow and broad population into additional subclasses that can be differentiated by firing and waveform features. In this regard, Ardid and colleagues (2015) recording in ventromedial and lateral prefrontal cortex and anterior cingulate cortex (ACC) (Brodmann areas 8, 9, 46) reported the identification of 7 distinct classes (4 broad and 3 narrow classes), a further study with the same traced 8 classes (5 broad and 3 narrow classes) in lateral

prefrontal (area 8) and ACC (Banaie Boroujeni, et al., 2021). Although interesting, nevertheless such a high number of classes would dramatically reduce the cell sample size for each class to be used for subsequent analyses, particularly for the narrow population, which accounts for 22.8 % of our whole sample in line with other studies in the prefrontal, but also potentially for the broad population given the large number of classes found in the literature (as can already be observed in the number of cells in each class in articles cited). One of the advantages of the dataset used in this study is a large enough sample of cells recorded for each task, however even in this case, the large reduction of the sample concerning the classification of cell subtypes would impact the reliability of the analysis for the quantification of coding magnitude and coding schemes. Both analyses are dependent on the sample size analysed, and cases of small sample size may impact the analysis and its reliability, e.g. specific neurons that, although not representative of the sample, may dominate the population analysis. Finally, although several subtypes of interneurons (e.g. those expressing Parvalbumin and Somatostatin interneurons) have been detected, most studies come from rodents. Currently, the monkey studies on these subclasses are limited (Banaie Boroujeni et al., 2021). They would require further recording approaches (e.g. optogenetic, juxtacellular recording) to ensure a reliable association between cell types and identified classes. However, we agree with the reviewer that investigating possible specific properties for cell subtypes is a potential interest for the scientific community. We have therefore added the following text in the newly expanded section titled "*Interpretational issues and limitations of the study*" in the discussion: "A second limitation of our study concerns the classification of putative interneurons as a unique class of cells. Recent studies have classified the major BW and NW classes into additional subclasses suitable for identifying subpopulations of separate types of interneurons and pyramidal cells^{38,47,54,91}. The challenge for future studies, using those methodologies, requires linking these functional subclasses to morphological classes of cell types. Such effort might lead to further distinguishing the coding schemes and coding magnitude specific to different cell subtypes, and enriching the characterization of the prefrontal microcircuitry's coding properties."

The authors should explicitly explain how the explained variance is computed in the "Strategy Response" (Fig. 3C). As far as I understood, the authors calculated the amount of variability explained by the fact it is a repeat-stay or change trial, but it seems not clearly stated in the paper.

We thank the reviewer for pointing out this potential misunderstanding. To make it immediately explicit to the reader how trials were selected for all variables of interest, we have now added this part at the beginning of the neural results: "For the response and the stimulus in the Strategy task, we selected the correct trials in which the response corresponded to one of the three possible correct target positions (top, right, left) and the three stimuli displayed, respectively. For novel and familiar S-R associations in the NovelMap and FamMap tasks, we analyzed only correct trials where each of the three stimuli was associated with only one response following the S-R mappings defined by the specific task rule."

Could the author provide information regarding the depth of electrode insertion?

We kept track of the individual electrode depths during the recordings. The plots below show the electrode depths in the populations classified as NW and BW populations. The depth was calculated using as 0 reference the depth corresponding to the change of activity associated with each electrode's entrance into the cortex. Unfortunately, it was not possible to accurately estimate the cortical layers matching the reported depths, using non laminar electrodes. We would prefer not to include these data in the manuscript for these reasons.

Rebuttal Figure 7. Depths of electrodes for cell types. Histograms show the percentage of broad (left) and narrow (right) cells recorded at varying electrode depths

It is unclear why the acronyms NR and BR were chosen for the narrow and broad spiking cells. I suggest using the more commonly used acronym NW and BW.

As suggested, we changed NR and BR to NW and BW to be consistent with the acronyms already used in the literature.

In plots A and B from Fig.2, it seems that the latency in the coding of the associations decreases in the FamTask compared to the NovelMap task in the NR population. It would be interesting to mention it and quantify such latency difference.

We are grateful to the reviewer for suggesting this analysis, which allowed us to identify a cell type-specific effect in the selectivity latencies between NovelMap and FamMap. We found that only the NW population anticipated its selectivity in the FamMap familiar context compared with NovelMap, but the BW population did not. To describe how the analysis was performed, we added this text to the methods: *"To investigate the selectivity latencies between cell types populations and associative tasks, we identified the first significant bin from the IS onset for each variable of interest and each cell."*

In addition to the comparison described earlier, we also evaluated the latency effect between cell types within the same variable to address comments from the other reviewers. The analysis is reported in the results as follows: *"To test for a difference in the selectivity latencies, we identified the first significant time bin within the IS period in the two cell type populations. We found that NW population coded significantly earlier, both novel (NovelMap, mean onset time, from IS onset: NW 233.0 ms; BW 281.1 ms, Kruskal-Wallis between time onset distributions, $p < 0.05$) and familiar associations (FamMap, mean onset time, from IS onset: NW 179.1 ms; BW 265.4 ms, Kruskal-Wallis between time onset distributions, $p < 0.05$). Moreover, we found a significant decrease in latencies of familiar versus new associations coding, in the NW population (Kruskal-Wallis between NW time onset tasks distributions, from IS onset, $p < 0.05$), but not in the BW population (Kruskal-Wallis between BW time onset tasks distributions, from IS onset, $p > 0.05$)."*

"Latency analyses revealed a tendency in the NW population to encode both the response (Strategy response, mean onset time, from IS onset: NW 289.3 ms; BW 327.1 ms, Kruskal-Wallis between time onset distributions, $p > 0.05$) and the stimulus (Strategy stimulus, mean onset time,

from IS onset: NW 229.2 ms; BW 283.5 ms, Kruskal-Wallis between time onset distributions, $p > 0.05$) earlier in the Strategy task, when compared to the BW population, however without achieving statistical significance.”.

Bibliography:

- Ardid, S. *et al.* Mapping of Functionally Characterized Cell Classes onto Canonical Circuit Operations in Primate Prefrontal Cortex. *J. Neurosci.* **35**, 2975–2991 (2015).
- Asaad, W. F., Rainer, G. & Miller, E. K. Neural Activity in the Primate Prefrontal Cortex during Associative Learning. *Neuron* **21**, 1399–1407 (1998).
- Asaad, W. F., Rainer, G. & Miller, E. K. Task-Specific Neural Activity in the Primate Prefrontal Cortex. *Journal of Neurophysiology* **84**, 451–459 (2000).
- Banaie Boroujeni, K., Tiesinga, P. & Womelsdorf, T. Interneuron-specific gamma synchronization indexes cue uncertainty and prediction errors in lateral prefrontal and anterior cingulate cortex. *eLife* **10**, e69111 (2021).
- Buschman, T. J., Siegel, M., Roy, J. E. & Miller, E. K. Neural substrates of cognitive capacity limitations. *Proc. Natl. Acad. Sci. U.S.A.* **108**, 11252–11255 (2011).
- Cromer, J. A., Machon, M. & Miller, E. K. Rapid Association Learning in the Primate Prefrontal Cortex in the Absence of Behavioral Reversals. *Journal of Cognitive Neuroscience* **23**, 1823–1828 (2011).
- Miller, E. K., Erickson, C. A. & Desimone, R. Neural Mechanisms of Visual Working Memory in Prefrontal Cortex of the Macaque. *J. Neurosci.* **16**, 5154–5167 (1996).
- Kadohisa, M. *et al.* Frontal and temporal coding dynamics in successive steps of complex behavior. *Neuron* **111**, 430-443.e3 (2023).
- Tsujimoto, S., Genovesio, A. & Wise, S. P. Neuronal Activity during a Cued Strategy Task: Comparison of Dorsolateral, Orbital, and Polar Prefrontal Cortex. *J. Neurosci.* **32**, 11017–11031 (2012).

- Wasmuht, D. F., Spaak, E., Buschman, T. J., Miller, E. K. & Stokes, M. G. Intrinsic neuronal dynamics predict distinct functional roles during working memory. *Nat Commun* **9**, 3499 (2018).

REVIEWERS' COMMENTS

Reviewer #1 (Remarks to the Author):

The authors have addressed all the comments properly and thoroughly, so I have nothing further to add other than a very minor oversight:

P6: "We found that overall the BW and NW task-related populations had ω^2 values and a higher coding magnitude in the NW population, comparable to the results in Fig. 3 (Supplementary Fig. 4a, d, g, j)." Had 'higher' ω^2 values

Reviewer #2 (Remarks to the Author):

The authors constructively addressed all major concerns and added important additional behavioral learning and neural decoding results to the manuscript that significantly broadens the scope and potential impact of the paper.

It is acknowledged and appreciated that considerable additional analysis was necessary for the revisions. The approaches, new results and figures are of highest quality and well interpretable.

There are two aspects that the author may want to check.

- The authors added a sentence in which the word 'already' seems unclear or imprecise. Could this be checked?:

"...that the prefrontal microcircuitry shifted toward a purely static scheme already after the learning was completed, which persisted when the associations became familiar across days, as seen in the FamMap".

Do the authors mean that a static coding schema was evident "...as soon as learning was completed", i.e. that is the stability was reflecting the completion of learning?

- The new result shows that narrow spiking neurons have stronger encoding than broad spiking neurons during the actual learning period. This is an exciting finding in the context of the study's task. The

authors added a brief discussion of this finding, but do not discuss how the finding of stronger narrow spiking neural coding related to previous findings in the literature. There are at least two recent studies of narrow spiking neurons in the prefrontal cortex during learning tasks that support the reported findings and provide converging evidence for the special role of narrow spiking neurons to have stronger learning related information in their firing rates:

- Banaie Boroujeni et al. (2021). Interneuron Specific Gamma Synchronization Indexes Cue Uncertainty and Prediction Errors in Lateral Prefrontal and Anterior Cingulate Cortex. eLife.
- Oemisch et al., (2019) Feature-specific prediction errors and surprise across macaque fronto-striatal circuits. Nature Communications.

It is fully at the discretion of the authors to discuss their findings with regard to the stronger cue- and outcome- related encoding of learning of narrow spiking neurons that these studies have found. As a reviewer I do not intend to impose references, but rather want to make aware of converging findings in the literature.

Reviewer #3 (Remarks to the Author):

I thank the authors for the careful and accurate answers, I am satisfied with how the questions were addressed.

I have a few comments:

Line 74: what is the difference between high and fast firing rates?

Line 75: I find the sentence "and cell types in information processing" superfluous.

Line 83: I find this sentence unclear "Our previous recordings". The author could write previously acquired datasets.

Line 56: if possible I would explain first the NovelMap, then FamMap, and lastly the Strategy task as it's the order in which are mentioned above.

Line 256: Sorted is clearer than classified

Line 273: The sentence "by removing those cells with a very low firing rate" is redundant, I would remove it.

Line 366: Don't repeat the word population

I believe that when the authors write cell types means distinct (or different) cell types (in lines 76, 80, 208..).

I found the sentence in line 270 not clear, consider re-writing: We found that overall the BW and NW task-related populations had ω_2 values and a higher coding magnitude in the NW population, comparable to the results in Fig. 3.

Discuss the possible molecular classification of the NW group, which may at least partially overlap with the Parvalbumin interneuronal population (as shown in mice by Kim et al., 2016).

Kim et al. also describe a specialized role of Parvalbumin interneurons in the Prefrontal cortex for attentional processing, potentially explaining the stronger recruitment observed in this neuronal population during the task (as discussed by the authors).

Supplementary Figure 5, wrong color code used.

Reviewer #1 (Remarks to the Author):

The authors have addressed all the comments properly and thoroughly, so I have nothing further to add other than a very minor oversight:

We thank the reviewer for the favorable consideration of our work.

P6: "We found that overall the BW and NW task-related populations had ω_2 values and a higher coding magnitude in the NW population, comparable to the results in Fig. 3 (Supplementary Fig. 4a, d, g, j)."

Had 'higher' ω_2 values

We thank the reviewer for spotting this mistake. We have changed the sentence as suggested: "We found that overall the BW and NW task-related populations had higher ω_2 values and a higher coding magnitude in the NW population, comparable to the results in Fig. 3".

Reviewer #2 (Remarks to the Author):

The authors constructively addressed all major concerns and added important additional behavioral learning and neural decoding results to the manuscript that significantly broadens the scope and potential impact of the paper.

It is acknowledged and appreciated that considerable additional analysis was necessary for the revisions. The approaches, new results and figures are of highest quality and well interpretable.

We thank the reviewer for his suggestions that helped to improve our manuscript.

There are two aspects that the author may want to check.

- The authors added a sentence in which the word 'already' seems unclear or imprecise. Could this be checked?:

"...that the prefrontal microcircuitry shifted toward a purely static scheme already after the learning was completed, which persisted when the associations became familiar across days, as seen in the FamMap". Do the authors mean that a static coding scheme was evident "...as soon as learning was completed", i.e. that is the stability was reflecting the completion of learning?

We thank the reviewer for pointing out that the sentence was unclear. As interpreted correctly by the reviewer, we meant that the static coding scheme occurred as soon as the learning was accomplished.

We have rephrased the sentence to make it clearer as follows: "These results suggest that the prefrontal microcircuitry shifted toward a purely static scheme as soon as the learning was completed, and that such static scheme persisted when the associations became familiar across days, as seen in the FamMap"

- The new result shows that narrow spiking neurons have stronger encoding than broad spiking neurons during the actual learning period. This is an exciting finding in the context of the study's task. The authors added a brief discussion of this finding, but do not discuss how the finding of stronger narrow spiking neural coding related to previous findings in the literature. There are at least two recent studies of narrow spiking neurons in the prefrontal cortex during learning tasks that support the reported findings and provide converging evidence for the special role of narrow spiking neurons to have stronger learning related information in their firing rates:

- Banaie Boroujeni et al. (2021). Interneuron Specific Gamma Synchronization Indexes Cue Uncertainty and Prediction Errors in Lateral Prefrontal and Anterior Cingulate Cortex. eLife.
- Oemisch et al., (2019) Feature-specific prediction errors and surprise across macaque fronto-striatal circuits. Nature Communications.

It is fully at the discretion of the authors to discuss their findings with regard to the stronger cue- and outcome- related encoding of learning of narrow spiking neurons that these studies have found. As a reviewer I do not intend to impose references, but rather want to make aware of converging findings in the literature.

As already requested by the reviewer at the first rebuttal, we added (reference 90, 91) and discussed the major role of interneurons during learning of the associative task employed in these studies. We have further modified this part to emphasize the convergence of their results with ours, as follows: "However, recent studies in the lateral prefrontal cortex have demonstrated a prominent role of putative interneurons to encode information crucial for the flexible learning of new stimulus-reward contingencies in a color-based learning task. In this task, the monkeys, after the presentation of two stimuli, had to identify the movement direction for the stimulus associated with a specific color, which was changed in consecutive reversal blocks. Putative interneurons encoded the stimulus's color to be monitored and modulated their activity during the progression of learning of the new color-reward contingencies after reversals^{90,91}, suggesting a pronounced recruitment in learning periods, consistent with our evidence."

Reviewer #3 (Remarks to the Author):

I thank the authors for the careful and accurate answers, I am satisfied with how the questions were addressed.

We thank the reviewer for acknowledging our effort in the answers.

I have a few comments:

Line 74: what is the difference between high and fast firing rates?

We added "fast" to stress the connection with a name used in the literature (fast spiking) to describe putative interneuron cells in addition to "narrow spiking", which emphasizes the high frequency of discharge of these cells, the same for regular/low firing. We removed it for clarity.

Line 75: I find the sentence "and cell types in information processing" superfluous.

We removed this last part of the sentence.

Line 83: I find this sentence unclear "Our previous recordings". The author could write previously acquired datasets.

We have in accordance with the reviewer, changed the sentence as follows:

"Our previously acquired datasets in the PFC with a novel (NovelMap) mapping task, a familiar (FamMap) mapping task, and a Strategy task7 (Fig. 1a-b) are ideal for addressing the study of such coding properties of the local microcircuitry."

Line 56: if possible I would explain first the NovelMap, then FamMap, and lastly the Strategy task as it's the order in which are mentioned above.

We believed the reviewer was referring to the text added after the first rebuttal from line 156. We followed the reviewer's suggestion, reordering the explanation of the variables studied in each task as they were presented later in the results: *"For novel and familiar S-R associations in the NovelMap and FamMap tasks, we analyzed only correct trials where each of the three stimuli was associated with only one response following the S-R mappings defined by the specific task rule. Associations between stimuli and responses were not changed during the mapping sessions. Consequently, we did not distinguish the contribution of response and stimulus cell selectivity to the coding of the S-R associations. However, as shown later, we used the Strategy task to study the coding of different cell types for stimulus and response independently. For the response and the stimulus in the Strategy task, we selected the correct trials in which the response corresponded to one of the three possible correct target positions (top, right, left) and the three stimuli displayed, respectively"*.

Line 256: Sorted is clearer than classified

We changed "we classified neurons" to "we sorted neurons" in the manuscript.

Line 273: The sentence "by removing those cells with a very low firing rate" is redundant, I would remove it.

We removed this last piece from the sentence.

Line 366: Don't repeat the word population

We removed the typo from the sentence.

I believe that when the authors write cell types means distinct (or different) cell types (in lines 76, 80, 208..).

We added "distinct" to each "cell types" in the lines suggested by the reviewer.

I found the sentence in line 270 not clear, consider re-writing: We found that overall the BW and NW task-related populations had ω_2 values and a higher coding magnitude in the NW population, comparable to the results in Fig. 3.

We modified the sentence, as also suggested by reviewer 1, as follows: "We found that overall the BW and NW task-related populations had higher ω_2 values and a higher coding magnitude in the NW population, comparable to the results in Fig. 3".

Discuss the possible molecular classification of the NW group, which may at least partially overlap with the Parvalbumin interneuronal population (as shown in mice by Kim et al., 2016).

Kim et al. also describe a specialized role of Parvalbumin interneurons in the Prefrontal cortex for attentional processing, potentially explaining the stronger recruitment observed in this neuronal population during the task (as discussed by the authors).

We included in the discussion a reference to rodent studies showing a role in working memory and occurrence in the medial prefrontal cortex of the parvalbumin (PV) and somatostatin (SOM) expressing

subtypes of interneurons. As follows: *"In rodents, molecular classification of interneuron subtypes according to their specific expression of parvalbumin (PV) and somatostatin (SOM) revealed a major occurrence in the medial prefrontal cortex and a role in working memory of such subtypes. Such subtypes, particularly the PV and partly the SOM interneurons, showed high firing levels and narrow waveforms similar to the properties found in the interneurons' population reported in our study"*.

Supplementary Figure 5, wrong color code used.

We think the reviewer is referring to Supplementary Figure 10. We have corrected the color code, and we thank the reviewer.